# More than Three Decades of Bm86: What We Know and Where to Go

**DOI:** 10.3390/pathogens12091071

**Published:** 2023-08-22

**Authors:** Laura Jane Bishop, Christian Stutzer, Christine Maritz-Olivier

**Affiliations:** Department of Biochemistry, Genetics and Microbiology, Faculty of Natural and Agricultural Sciences, University of Pretoria, Pretoria 0083, South Africa; u14053790@tuks.co.za (L.J.B.); u04160169@tuks.co.za (C.S.)

**Keywords:** tick, vaccine, Bm86, antigen, vaccine development

## Abstract

Tick and tick-borne disease control have been a serious research focus for many decades. In a global climate of increasing acaricide resistance, host immunity against tick infestation has become a much-needed complementary strategy to common chemical control. From the earliest acquired resistance studies in small animal models to proof of concept in large production animals, it was the isolation, characterization, and final recombinant protein production of the midgut antigen Bm86 from the Australian cattle tick strain of *Rhipicephalus* (*Boophilus*) *microplus* (later reinstated as *R.* (*B.*) *australis*) that established tick subunit vaccines as a viable alternative in tick and tick-borne disease control. In the past 37 years, this antigen has spawned numerous tick subunit vaccines (either Bm86-based or novel), and though we are still describing its molecular structure and function, this antigen remains the gold standard for all tick vaccines. In this paper, advances in tick vaccine development over the past three decades are discussed alongside the development of biotechnology, where existing gaps and future directives in the field are highlighted.

## 1. Introduction

The burden of ticks and tick-borne diseases on both humans and animals has been chronicled by scholars (i.e., in art and writings) since ancient antiquity (Egypt, ca. 1550 BC.; Greece, ca. 355BC) [1,2]. Ticks are competent vectors for the widest variety of cellular parasites and pathogens [3]. The economic impact on the global cattle industry of the one-host Asiatic blue tick, *Rhipicephalus microplus*, alone has been conservatively estimated at around USD 13.9–18.7 billion per year [4] to as high as USD 22–30 billion per year [5]. Consequently, considerable effort and resources have been invested in developing practices and products to control tick infestations, as well as limit the spread of and/or treat their associated diseases. For the livestock industry, the most prevalent tick control method has been the application of chemical acaricides. However, from the first discovery of organochlorines (OCs) in the 1940s to the introduction of the growth inhibitor fluazuron (FL) (early 1990s) into the acaricide market, the first report of resistance to the acaricide actives follows almost like clockwork within 10 to 20 years from the first introduction [6] (Figure 1, Appendix A). Resistance to synthetic pyrethroids was published within as little as two years following their introduction into the market in 1977.

To alleviate the selective pressures placed on tick populations by chemical control, multiple approaches are needed within a sustainable integrated pest management program that can include: biological control (e.g., plant extracts [7], predators, parasitoids [8], and pathogens of ticks [9], genetic control (e.g., raising tick-resistant cattle breeds and sterile insect techniques) [10,11,12], and immunological control (e.g., tick vaccines) [13,14,15]. Immunological control of ticks and tick-borne diseases has a long history of research (see a special issue on classic papers in tick and tick-borne disease research at https://www.mdpi.com/journal/pathogens/special_issues/10thAnniversary_Tick; accessed 20 June 2023). One of the most challenging aspects of parasite vaccine development remains the identification of a protective antigen. The development of tick vaccines evolved in the wake of technological developments. Vaccines evolved from the first attenuated live vaccine for chicken cholera developed by Louis Pasteur in 1879 [16] to the first recombinant subunit vaccine against hepatitis B produced in 1986 using a yeast production host [17] (see Figure 1 and Appendix A). A subsequent seminal moment in tick vaccine research and development was the characterization and recombinant production of Bm86 by the researchers at the Commonwealth Scientific and Industrial Research Organization, or CSIRO (St Lucia QLD 4067, Australia) [18]. This paper seeks to highlight the milestones in recombinant subunit vaccine development pioneered by Rand et al. (1989) [19] using the best-described and most effective tick vaccine antigen identified from the cattle tick species *Rhipicephalus* (*B.*) *microplus*, known as Bm86.

We will discuss some of the prior history leading up to the seminal work carried out by Rand et al. [19] and then the subsequent development path of this subunit vaccine over the next almost four decades. The hard lessons learned from this antigen and its formulation into an effective vaccine provide a roadmap for all other consequent recombinant tick vaccines. The highlighted successes and failures, as well as the obstacles and unknowns, can provide valuable insights for others in this field and hopefully make a journey fraught with difficulties a “smoother” one. Please note that the Australian cattle tick (and the Yeerongpilly vaccine strain) was previously classified as *Boophilus microplus*, then later as *Rhipicephalus* (*B.*) *microplus*, but in 2012 this strain was reinstated as a separate tick species, i.e., *R.* (*B.*) *australis* [20], and will hence forth be addressed as such in the rest of this manuscript.

## 2. The Tick Vaccine Movement (Pre-1980s): Laying the Groundwork for Proof-of-Concept

In 1918, Johnston and Bancroft investigated apparent resistance to *R.* (*B.*) *australis* infestations on Australian farms in New South Wales [21] and Queensland [22]. It was reported that some cattle breeds appeared to be naturally more resistant, while other breeds could develop resistance following repeated exposure, to an extent that was dependent on the nutritional state and overall health of the individual animal [22]. Similar so-called acquired tick resistance (ATR) would be described in the 1970s for *Bos taurus*, *Bos indicus,* and crossbred cattle repeatedly infested with *Amblyomma americanum* [23], *Ixodes holocyclus* [24], and *Haemaphysalis longicornis* [25]. As part of the parasite-host interaction, ticks inject their saliva, which contains numerous bioactive molecules, to counteract and/or suppress the host’s hemostatic and immunological responses (including inflammation) to enable feeding [26]. In response, host defence pathways are activated (innate and acquired immunity) to overcome the ticks’ assault and impair attachment, feeding, and reproduction [27]. Acquired tick resistance has also been described in several other host species, such as pigs, rabbits, guinea pigs, and mice, following a single or repeated infestation(s) (depending on host animal and tick species) [28]. Moreover, it has been suggested that aside from other morphological and physiological traits, immunobiological effector cells such as skin-resident memory T cells are responsible for recruitment of inflammatory basophils to the skin lesion to facilitate ATR [28]. In 1976, Roberts and Kerr observed that a degree of passive immunity could be obtained in cattle immunised with hyperimmune serum isolated from donor cattle that were repeatedly infested with *R.* (*B.*) *australis* [29]. This indicated (to some degree) the involvement of humoral responses to ATR. Even today, researchers are still investigating various host traits (including genetic and immunological markers) that confer natural resistance to tick infestation in cattle [30,31,32]. However, animals never display sterilising or neutralising immunity to tick infestations because of ATR. Moreover, crossbreeding animals to obtain natural ATR traits can have negative production tradeoffs which tend to be undesirable for commercial farming [33].

Vaccines, even ones with limited efficacy, can be included to increase the success of an integrated tick and tick-borne disease management program to protect both humans and animals against infection/infestation [34,35]. In 1939, the seminal work performed in guinea pigs by Dr. William Trager demonstrated the first proof of concept [36]. Protection against infestation was conferred in guinea pig hosts following immunisation with *Dermacentor varabilis* larval extracts, which could be passively transferred to naive recipients, resulting in ~50% reduction in larval burdens [36]. The latter showed that protection was in part driven by antigen-specific antibodies and the findings of this study paved the way for the eventual development of tick vaccines that: reduces tick infestations (as well as the prevalence of associated tick-borne diseases), impairs tick feeding and nutrient acquisition, as well as reproduction and fecundity of the offspring. Trager’s observations were further confirmed in other tick species (e.g., *Dermacentor andersoni* and *Heamaphysalis leporis palustris*) on small mammalian hosts (e.g., deer, mouse, and rabbits) [36,37].

In the case of cattle ticks, Roberts opined in 1968 that based on the variability of ATR in cattle hosts, the feasibility of a vaccine against *R.* (*B.*) *microplus* would be limited to the innate capability of any host animal to develop natural immunity [38]. However, in 1975, it was suggested that targeting specific novel antigens (e.g., moulting hormones) for host humoral responses could provide protection against tick infestation [39]. Consequently, the seminal work performed by Allen and Humphreys in the 1970s [40] ushered in the era of tick vaccine development for production animals [41]. Following pilot immunisation trials in guinea pigs, Allen and Humphreys immunised Hereford-crossbred calves with antigen prepared from *D. andersoni* midgut and reproductive tissues formulated in Freund’s incomplete adjuvant [40]. Significant reductions of around 80% in female weights, egg, and viable larvae produced were observed, and it was suggested that such an approach would be feasible for studies in cattle infested with *R.* (*B.*) *australis* [41].

## 3. The Dawn of Cattle Tick Subunit Vaccines (1980s to 1990s): The Discovery of Bm86

The 1980s ushered in a new direction for tick control as investigators from CSIRO (Queensland, Australia) took the first steps towards the development of cattle tick subunit vaccines. Bovine host responses against infestations with the cattle tick *R.* (*B.*) *australis* were conducted [42,43]. Consequently, Johnston and colleagues [43] (1986) found that immunising cattle with extracts from whole adult *R.* (*B.*) *australis* females produced antibody-mediated immune responses, resulting in significant reductions in fed female weights (around 67% and 77%) and the number of dropped engorged females (around 62% and 72%) in vaccinated animals relative to non-vaccinated controls. So-called red ticks were observed during these trials, and further histological studies conducted by Agdebe and Kemp (1986) [42] confirmed damage to the tick midguts that fed on vaccinated animals, characterised by a delayed development of the midgut tissues where type 2 secretory cells were budding off into the ruptured lumen and the tick haemolymph was packed with host red blood cells, leukocytes, and other lysed tissue products. Specifically, the development of third generation digest cells (subtype I) appeared to be inhibited, while subtype II digest and basophilic cells were absent or unidentifiable [42]. Due to the frequency of ruptured midgut epithelial cells observed in ticks fed on vaccinated animals, it was suggested that the prominent protective antigens were located on the luminal surfaces of the plasma membranes of these cells. Kemp *et al*. (1989) [44] further showed that the tissue damage was observed in adult (both male and female) ticks. Additionally, in vitro feeding assays using whole blood or serum, as well as heat-inactivated serum (to remove complement), demonstrated that both antibody- and complement-mediated responses (at least in part) are responsible for this damage.

It was the studies published by Willadsen and colleagues in the latter half of the 1980s [45,46], that finally described the protective antigen, i.e., a 89 kDa midgut glycoprotein designated as Bm86 that induced immunoprotection during cattle infestation trials. The antigen was named according to the year 1986, in which the tick protein was discovered and its protein sequence was partially characterized [47]. Classical protein fractionation protocols were used to isolate and purify the native protective antigen [48]. About 185 μg of native Bm86 protein was purified from 31.2 g of crude extract prepared from some 50,000 adult female ticks. The purified protein (2.3 or 17 μg per injection) was formulated in Freund’s complete and incomplete adjuvant for three injections, which resulted in 65% reduced engorgement, 33% reduced engorgement weight, more than 50% reduction in egg production, and 92% reduction in larval numbers in cattle infestation trials [45]. The predicted amino acid sequence of the unprocessed Bm86 protein was estimated to be 650 amino acids in length [19], containing: a signal peptide for export (1–19 amino acids), several glycosylation sites, a single C-terminal transmembrane section, which is replaced in the mature protein by a glycosyl-phosphatidylinositol or GPI membrane anchor [45], several Epidermal Growth Factor (EGF)-like domain repeats, as well as approximately 66 cysteine residues [19,49]. The EGF-like domain repeats were proposed to be involved in blood coagulation and cell growth [49]. The high cysteine content was expected for cell surface proteins, and the cysteine clustering was indicative of intramolecular disulphide bridging [49]. Histology studies using Bm86-specific antisera indicated that this protein is indeed located on the extracellular surface of digest cells in the tick gut [50].

Due to the availability of the protein sequence and ease of production, the first recombinant version of Bm86, conjugated to a β-galactosidase amino-terminal tag, was produced in *Escherichia coli* inclusion bodies [19]. This construct finally resulted in an estimated 89% reduction in the reproductive capacity of *R.* (*B.*) *australis* in the first controlled bovine vaccination and infestation trials [49]. This demonstrated the advantage of tick vaccination with recombinant proteins over vaccination with fractionated native proteins [45]. Initial studies suggested that the protective effect between native Bm86 and recombinant Bm86 produced in *E. coli* appeared to be determined only by the protein itself and that secondary determinants such as glycosylation did not significantly contribute to protection [51]. However, cattle infestation trials testing various truncations of Bm86, with or without the β-galactosidase amino-terminal tag, showed that the bacterially produced protein was a far poorer antigen than native Bm86 [49]. It was concluded that the high response achieved with the bacterially produced protein was mainly attributed to the β-galactosidase tag. Subsequent eukaryotic protein production studies in *Aspergillus nidulans* or *A. niger* [52] indicated that these platforms were not effective in producing enough recombinant protein for pilot vaccinations. Bm86 produced in baculovirus-transformed insect cells afforded higher efficacy (i.e., a 91% reduction in reproductive capacity) for the control of tick infestations than Bm86 produced in bacteria [49,53]. In contrast to findings from Bm86 produced in *E.coli*, this suggests that antigen efficacy is dependent on correct protein folding and glycosylation, which together contribute towards protein stability and longevity of the immune response [53]. Incidentally, Richardson et al. (1993) [53] also identified a 160 kDa immuno-reactive band during SDS-PAGE analysis that indicated the possibility of Bm86 occurring as a disulphide-linked dimer.

Field trials started in 1989 and were performed in Australia [54], Cuba [55], and Brazil [56] (Table 1). The Australian Bm86-based vaccine was finally patented and registered for commercial sale in 1994 via Biotech Australia Pty. Ltd. (Patent number: WO 95/04827) [54,57]. The vaccine TickGARD^®^ consisted of the Bm86 protein (Yeerongpilly strain) produced in bacteria and formulated in Montanide^TM^ (Seppic, Courbevoie, France) [54]. This formulation was later adapted in 1996 to include an antigen produced in the yeast *Pichia pastoris* and Vaximax (Marcol25: Montanide^TM^ 888 in a 9:1 ratio) (Seppic, Courbevoie, France) as an adjuvant that was registered as TickGARD^®^ Plus [54].

During the same period, in Latin America, the Bm86 antigen was amplified from a Cuban stock of the Yeerongpilly strain (i.e., Australian isolate) of *R.* (*B.*) *australis* and produced recombinantly in the yeast *Pichia pastoris* [58]. Formulations in Freund’s complete and incomplete adjuvant of this antigen were tested in cattle vaccination and tick infestation studies and resulted in a 70% reduction in tick reproductive ability relative to negative controls [58]. Consequently, the commercial vaccines Gavac^®^ and later Gavac Plus^®^ (Herber Biotec S.A., CIGB, Camagüey, Cuba) were developed and remain commercially available today. Like TickGARD^®^, Gavac^®^ was formulated in Montanide^TM^ 888 [55,59]. The glycosylated recombinant Bm86 produced in yeast would later prove to have superior immunogenicity over the non-glycosylated Bm86 produced in bacteria [47,60,61].

To improve the vaccine formulation, the first Bm86-based combination vaccine was tested in cattle trials with the inclusion of Bm91 [62], an angiotensin-converting enzyme-like protein [63]. Bm91 was identified and characterised by Riding et al. (1994) [64] as a low-abundance glycoprotein of the salivary glands and midgut of *R.* (*B.*) *australis*. Inclusion of this antigen did significantly improve the vaccine efficacy [62]; however, it was not included in later formulations of TickGARD^®^. Almost two decades later, Bm91 was cloned from a Thai strain of *R. microplus* and produced recombinantly in the yeast *Pichia pastoris* [65]. However, it would prove a poor vaccine candidate as only a 6% and 8% reduction in the reproductive efficiency index and egg viability were demonstrated, relative to control animals, following cattle vaccination and infestation trials [65].

The cost-effectiveness of vaccination with Gavac^®^ was demonstrated in a Cuban study using more than 260,000 cattle that showed a 60% reduction in the number of acaricide treatments relative to non-vaccinated controls over the study period. Moreover, the additional reduction in tick loads, as well as transmission of babesiosis, resulted in calculated savings of 23.4 USD per animal per year [66]. In Columbia, cost savings of $3 per animal per year were estimated [67]. It was ultimately demonstrated that Bm86-based vaccination has several potential benefits in tick control, including: (1) cost-effectiveness; (2) reduction in environmental contamination via a reduction in chemical acaricide use; (3) reduction in the rise of acaricide-resistant tick populations via a reduction in chemical acaricide use; and (4) an indirect reduction in the transmission of TBDs, either by reducing tick populations and/or by affecting the capacity of the tick as a vector [61,68,69].

The full molecular function of the antigen and the immune profile elicited by Bm86-based subunit vaccines remain largely uncharacterized (see Table 1). However, responses almost mirror those determined from crude extract vaccinations, which evidences that the main antigen responsible for the protective responses was indeed Bm86. Briefly, early evidence from the 1990s pointed towards a type 2 adaptive immune response, typified by a strong humoral response [19,44,45,68]. A strong total IgG anamnestic response is also observed [56,58,59,60,68,70], with a few studies reporting switching to an IgG1 isotype [68,71,72,73], all of which indicate good memory B cell activation following vaccination [68].

**Table 1 pathogens-12-01071-t001:** A summary of Bm86-based vaccines that have been evaluated in bovine vaccination trials achieving an efficacy of 50% or above. Vaccines are listed in chronological order within categories of: Bm86-based recombinant antigen vaccines (indicated in blue), Bm86-based epitope vaccines (indicated in orange), and combination vaccines with Bm86 (indicated in green).

Vaccine and Major Antigen(s)	Breed	Adjuvant	Vaccination Regime	Efficacy in Bovine Vaccination Trials	Immune Markers	Reference(s)
**Bm86-Based Recombinant Antigen Vaccines**						
**Bm86** TickGARD^®^ and TickGARD^®^ Plus Discontinued Commercial Vaccines (Intervet Pty. Ltd., Bendigo, Australia)	-Hereford (*Bos taurus taurus*) [19,54,74] -Holstein Friesian (*Bos taurus taurus*) [60]	-Freund’s complete adjuvant [19,45] -Montanide^TM^ (TickGARD^®^) [54] -Vaximax i.e., Marcol25: Montanide 888 (9:1) (TickGARD Plus^®^) [57,60]	Dosage: -2.3 μg (trial 1); 17 μg (trial 2) [18,45] -50 μg [51,61] Regime: -Three vaccinations at weeks 0, 4, and 8 [18,45] -Three vaccinations at weeks 0, 7, and 17; or two vaccinations at weeks 0 and 7 [51] -Three vaccinations at weeks 0, 4, and 9 [61]	REF: 20–56% Total efficacy: Ranges from 0 of >90%	Serum antibody levels: Anti-Bm86 mean IgG titres (dilution 1:4000): [60] -Day of 1st injection: 187. -2 weeks after 1st injection: 646. -2 weeks after 2nd injection: 2633. -8 weeks after 3rd injection: 582.	Patents: [75,76] Literature: [19,45,54,57,60]
**Bm86** Gavac^®^ Commercially available vaccine (Heber Biotec S.A., Cuba)	-Holstein Friesian (*Bos taurus taurus*) [58,70,73] -Crossbred (*Bos taurus × Bos indicus*) [55,77] *-Bos taurus*, *Bos indicus* and crossbred dairy and beef cattle (*Bos taurus* × *Bos indicus*) [56,68]	-Freund’s complete (1st vaccination) and incomplete (2nd and 3rd vaccination) adjuvant [58] -Montanide^TM^ 888 VG (Gavac^®^) [55,56,59,68,70,73,77] -Saponin (saponin white pure, Merck [73]	Dosage: -100 μg [52,53,59,68,70,71] -400 μg [59] -50 μg [66] Regime: -Three vaccinations at weeks 0, 4, and 7 [52,53,59,68,70] -Three vaccinations at weeks 0, 4, and 7 followed in field trials by a maintenance dose approximately every 6 months [66] -Three vaccinations at weeks 1, 3, and 7 [71]	REF: 9–100% Total efficacy: Ranges from 0 to ≤100%	Serum antibody levels: Anti-Bm86 mean IgG titre: -2 weeks after 3rd injection (dilution n.i.): 100 × 10^4^ [58] -2 weeks after 3rd injection (dilution n.i.): 48,000 ± 3500 [55] -3 weeks after 2nd injection (dilution n.i.): 16,000 [56] -18 weeks after 3rd injection (dilution n.i.): 6000 [56] -Average of 9 pen trials 2 weeks after 3rd injection (dilution n.i.): 6351 ± 578 [59] -2 weeks after 2nd injection (dilution 1:320): ±8000 [68] -5 weeks after 3rd injection (dilution 1:640): 7000 ± 1000 [70] Mean titres 2 weeks after 3rd injection (dilution n.i.): [73] Bm86 and Montanide^TM^ 888 VG (Gavac^®^) -Total IgG: 23,744. -IgG1: 11,763 -IgG2: 1114 Bm86 and Saponin -Total IgG: 24,639 -IgG1: 8611 -IgG2: 2560 Anti-Bm86 mean OD values 2 weeks after 3rd injection (dilution 1: 1000): ± 0.3. [77]	Patent: [78] Literature: [55,56,58,59,68,70,73,77]
**Bm95** (Bm86 strain variant) Gavac^®^ Plus Commercially available vaccine (Heber Biotec S.A.,Cuba)	-Holstein Friesian (*Bos taurus taurus*) [70] -Crossbreed (n.i.) [79] -Crossbreed (*Bos taurus × Bos indicus*) [80]	-Montanide^TM^ 888 [70,80] -Algel (Indian Immunological Ltd., Hyderabad) [79]	Dosage: -100 μg [68,73] -200 μg [72] Regime: -Three vaccinations at weeks 0, 4, and 7 [68] -Four vaccinations at weeks 4, 8, 12 and 16 [72] -Three vaccinations at weeks 0, 4, and 7; or two vaccinations at weeks 0 and 4; or at weeks 0 and 7 [73]	Total efficacy: -58–89% [70] -81.27% [79] Tick biological parameters: Significant reduction in weight of engorging females, weight of eggs and egg viability between vaccinated and unvaccinated groups. No significant differences observed between different vaccination schemes [80]	Serum antibody levels: Mean IgG titre 2 weeks after 2nd injection (dilution 1: 320): [70] -Anti-Bm86 ± 8000. -Anti-Bm95 ± 9000. Anti-Bm95 mean IgG titre 3 weeks after 3rd injection (dilution n.i.): [79] -7979.9 ± 312.5. Anti-Bm86 mean IgG titres (dilution n.i.): [80] Bovines vaccinated at weeks 0 and 4: -8 weeks after 2nd injection: 6000 ± 1000. -3 weeks after re-vaccination at week 24: 4000 ± 200. Bovines vaccinated at weeks 0 and 7: -5 weeks after 2nd injection: 5000 ± 100. -3 weeks after re-vaccination at week 24: 6500 ± 1000. Bovines vaccinated at weeks 0, 4, and 7: -5 weeks after 3rd injection: 7800 ± 1000. -3 weeks after re-vaccination at week 24: 6000 ± 1000.	Patent: [81,82] Literature: [70,79,80]
**Bm86** Mozambique strain	Crossbreed (n.i.)	Montanide ISA 50V (Seppic, France)	Dosage: -100 μg Regime: -Three vaccinations at weeks 1, 3, and 7	Total efficacy: 70.4	Serum antibody levels: Anti-Bm86 mean OD value for IgG 2 weeks after 3rd injection (dilution 1: 1000): ±1.18.	Literature: [77]
**Bm86-Based Epitope Vaccines**						
**SBm4912®** Bm86 SYNTHETIC EPITOPE CONSTRUCT	Jersey-breed (*Bos Taurus*)	Saponin	Dosage: -2 mg Regime: -Three vaccinations at weeks 0, 4, and 8	Total efficacy: 72.4%	Serum antibody levels: Anti-Bm86 mean OD value for IgG 4 weeks after 3rd injection (dilution n.i.): 1.2 ± 0.4.	Patent: [83] Literature: [84]
**SBm7462®** Bm86 synthetic epitope construct	Holstein Friesian (*Bos taurus taurus*)	Saponin	Dosage: -2 mg Regime: -Three vaccinations at weeks 0, 4, and 8	Total efficacy: 81.05%	Serum antibody levels: Anti-Bm86 mean OD value for 2 weeks after 3rd injection (dilution 1: 400): IgG: 1.2 ± 0.2. IgG1: 1.0 ± 0.2. Histology of LN tissue: -Increased GCs in B-cell follicles. -Hyperplasia of medullary cord. -Presence of antigens in APCs. -Apoptosis. Phenotype of circulating PBMCs: -Increased B-cells (CD21+). -Variation in WC1+ γδ T cells (CC101+).	Patent: [83] Literature: [72,84]
**rSBm7462^®^** Recombinant peptide derivative of **SB**m7462®	Holstein Friesian (*Bos taurus taurus*)	Saponin	Dosage: -2 mg Regime: -Three vaccinations at weeks 0, 4, and 8	Total efficacy: 72.4%	Serum antibody levels: -IgG mean OD value 1 week after 3rd injection (dilution 1: 100): 2.3 ± 0.5). Cytokine gene expression: -Upregulated: IL-4, INF-ƴ, IL-10, and IL-12. -Down-regulated: TNF-α. Histology of LN tissue: -Increased GCs in B-cell follicles. -Hyperplasia of medullary cords.	Patent: [85,86] Literature: [87]
**Combination Vaccines with Bm86**						
**Bm95-MSP1a** Recombinant Bm95 + recombinant *Anaplasma marginale* MSP1a	Beefmaster (*Bos taurus indicus*) × Charolais (*Bos taurus taurus*)	Montanide ISA 50V2 (Seppic, France)	Dosage: -120 μg Regime: -Three vaccinations at weeks 0, 4, and 8	Total efficacy: 64%	Serum antibody levels: IgG mean OD value 2 weeks after 3rd injection (dilution 1: 100): 0.5 ± 0.1.	Literature: [88]
**Bm86 + SUB** Recombinant Bm86 and SUB (salivary gland) proteins	Hereford/Holstein (*Bos taurus taurus*) mixed breed	Water-in-oil emulsion: Saponin QuilA (water) in Montanide ISA 50 V2 (Seppic, France) (oil)	Dosage: -100 μg Regime: -Three vaccinations at weeks 0, 3, and 6	REF: 97%	Serum antibody levels: Mean IgG titre in 2Log 3 weeks after 3rd injection (dilution n.i.): Bm86 + SUB injected separately: -Bm86: 10.2. -SUB: <7. Bm86 + SUB dual vaccine: -Bm86: 10.6. -SUB: 10.0.	Patent: [89]
**pP0-Bm86** Synthetic peptide derivative of p0 ribosomal protein conjugated to recombinant Bm86	Cuban Siboney: 5/8 Holstein (*Bos taurus taurus*) 3/8 Zebu (*Bos taurus indicus*)	Montanide ISA 50V (Seppic, France)	Dosage: -500 μg Regime: -Three vaccinations at weeks 0, 3, and 5	Total efficacy: 84%	Serum antibody levels: IgG mean OD value 2 weeks after 3rd injection (dilution n.i.): -anti-pP0: ± 10,000. -anti-Bm86: ± 8000.	Literature: [90]
**pSUB + Bm86** SUB synthetic peptide derivative and recombinant Bm86	*Bos taurus* × *Bos indicus* and 5/8 Holstein	Montanide ISA 50 V (Seppic, France)	Dosage: -100 μg Regime: -Three vaccinations at weeks 0, 4, and 9	Total efficacy: Peptide SUB alone: 67% Bm86 alone: 56% Peptide SUB + Bm86: 49%	Serum antibody levels: IgG mean OD value 2 weeks after 3rd injection (dilution 1: 100): -anti-pSUB alone: 1.9 ± 0.1. -anti-Bm86 alone: 1.1 ± 0.1. -anti-pSUB dual: 1.7 ± 0.1. -anti-Bm86 dual: 1.4 ± 0.1.	Literature: [91]
**Bm86 + H. *anatolicum* SUB and tropomyosin (TPM)**	*Bos taurus* × *Bos indicus*	Montanide ISA 50 V2 (Seppic, France)	Dosage: -100 μg Regime: -Three vaccinations at weeks 0, 4, and 8	Total efficacy: 87.6%	Serum antibody levels: Anti-rBm86 OD values across 6 bovines 2 weeks after 3rd injection (dilution 1:16,000): -IgG: 0.68–0.75 (IgG titre 90 DPI: 1:102,400). -IgG1: 0.68–0.75. -IgG2: 0.25–0.3. Anti-rSUB OD values across 6 bovines 2 weeks after 3rd injection (dilution 1:16,000): -IgG: 0.65–0.85 (IgG titre 75 DPI: 1:64,000). -IgG1: 0.68–0.80. -IgG2: 0.36–0.68. Anti-rTPM OD values across 6 bovines 2 weeks after 3rd injection (dilution 1:8000) -IgG: 0.6–0.72 (IgG titre 90 DPI: 1:51,200) -IgG1: 0.62–0.8. -IgG2: 0.28–0.75.	Literature: [92]

Total efficacy = 100 [1 − (CRT × CRO × CRF)], where CRT = reduction in the number of adult female ticks; CRO = reduction in oviposition; CRF = reduction in egg fertility as compared to the control group. REF = 100 [1 − (NTV/NTC)], where REF = reduction in the number of engorged females; NTV = total number of engorged females in the vaccinated group; NTC = total number of engorged females in the control group. N.i. = not indicated. LN = lymph node. GC = germinal centre. IgG = immunoglobulin G. APC = antigen presenting cell. SUB = Subolesin. MSP = major surface protein. IL = interleukin. TNF = tumour necrosis factor. IFN = interferon. DPI = days post injection.

Ingested bovine antibodies can recognise and bind to the surface-expressed GPI-linked Bm86 membrane protein in the tick midgut [45,93], leading to (a) inhibition of the endocytic ability of tick gut cells and (b) lysis of the tick gut cells. This disrupts the lining of the gut and causes leakage of host blood into the tick’s body cavity, which ultimately prevents digestion and adequate nutrient uptake required for effective ovipositioning [19,44]. Tissue lysis could also be assisted by the involvement of the complement system; in vitro feeding assays conducted in 1989 by Kemp et al. [44] and a complement fixation test with serum from vaccinated cattle [71] provide some evidence for this. However, the latter is not a sensitive or reliable indicator of complement activation. Overall, vaccination with the Bm86 subunit vaccines led to a significant reduction in tick biological parameters that included tick survival, engorgement weight of females, capacity for ovipositioning, survival of progeny, reproductive potential in successive generations, and an overall decline in tick populations [19,44,55,58,68,74].

## 4. First Encounter with the Antigen Diversity-Driven Bottleneck (2000–2020): Geographical Variation, Cross-Species Protection, and Attempts to Improve Bm86-Based Vaccines

In the 2000s, more extensive cattle infestation trials were conducted in both pilot and field formats to evaluate the efficacy of the commercial Bm86 subunit vaccines, which proved to have high efficacy in the initial controlled trials [54,55,56,57,58]. Additionally, both commercial vaccines were also tested for potential cross-species protection [94] and efficacy in various geographic areas [68,95] in the previous decade (Table 1 and Table 2). This was to demonstrate “universality”, as well as open new global markets for the application of these vaccines in tick control.

Field trials using the commercial vaccines led to several reports of varying levels of vaccine efficacy observed for local tick strains in different geographic locations (see Table 1). Such trials conducted in the late 1990s with Gavac^®^ demonstrated a range of efficacy (51% to 100%) against various South American (including Argentina, Mexico, Colombia and, Brazil) strains of *R. microplus*, as well as the Australian (i.e., Yeerongpilly) isolate of *R.* (*B.*) *australis* [68] (Table 1). Studies conducted in the 2000s, showed that vaccine efficacy ranged from zero to almost 100% for both TickGARD^®^ [60] and Gavac^®^ vaccines [60,80] (Table 1). Ultimately, this variation in efficacy reduced the acceptance of Bm86-based vaccines globally despite the observed benefits.

To explain the observed variation in vaccine efficacy, one hypothesis proposed was that sequence heterogeneity in the Bm86 vaccine antigen is the contributing factor. Therefore, in the late 1990s, studies attempted to link the range in efficacy to variations within the transcribed sequences of Bm86 [95]. For the Australian *R.* (*B.*) *australis* strains, up to 2.8% in Bm86 sequence diversity did not result in significant differences in sensitivity to vaccination with the TickGARD^®^ antigen. This notion was supported by a more recent study looking at *R.* (*B.*) *australis* strains from New Caledonia that shared a ≥97.6% sequence identity with the Bm86 vaccine antigen isolated from the Yeerongpilly laboratory strain [96]. In the subsequent homologous vaccination and tick challenge trial, 74.2% vaccine efficacy was obtained using a New Caledonian isolate of the Bm86 vaccine antigen that was produced in a bacterial protein production host [96].

**Table 2 pathogens-12-01071-t002:** A summary of Bm86, Bm86 homolog, and Bm86 ortholog vaccines that have been evaluated in bovine vaccination trials against different tick species. Vaccines are listed in chronological order.

Vaccine and Major Antigen(s)	Breed	Adjuvant	Vaccination Regime	Tick Species	Efficacy in Bovine Vaccination Trials	Immune Markers	Reference(s)
**Cross-Species Testing of Bm86, Bm86 Homolog and Bm86 Ortholog Vaccines**							
**Bm86** Tick-GARD® and TickGARD^®^ Plus Discontinued commercial vaccines (Intervet Pty. Ltd., Bendigo, Australia.)	1. Holstein Friesian (*Bos taurus taurus*) [97] 2. N.i. [98]	Montanide ISA 50V (Seppic, France)	Dosage: -25 μg [97,98] Regime: -Two vaccinations at weeks 0 and 4 [97,98]	*1. R. annulatus* [97] 2A. *R. microplus* [98] 2B. *R. decoloratus* [98] 2C. *R. appendiculatus* [98] 2D. *H. a. anatolicum* [98] 2E. *H. dromedarii* [98] 2F. *A. variegatum* [98]	Total efficacy: 1. 100% [97] 2A. 74% [98] 2B. 70% [98] 2C. 0% [98] 2F. 0% [98] Reduction in tick parameters: 2D. 50% reduction in the total weight of engorging nymphs. [98] 2E. 95% reduction in number of nymphs; 55% reduction in weight of surviving ticks. [98]	Serum antibody levels: 1. Mean IgG titre after 2nd injection: [97] -30 days 1:1600–1:24,000. -90 days: 1:6400–1:24,000. -138 days: 1:6400. -160 days: 1:400 (after repeated infestation). -270 days: 1:400 (after repeated infestation). -390 days: 1:400–1600 (after repeated infestation). 2. Mean IgG titre after 2nd injection (dilution 1:500): n.i. [98]	Patents: [75,76] Literature: [97,98]
**Bm86** Gavac^®^ Commercially available vaccine (Heber Biotec S.A., Cuba)	1. Aberdeen Angus and cross-breed (n.i.) [94] 2. Crossbreed (n.i.) [77] 3. *Bos taurus* X *B. indicus* [99] 4. Holstein Friesian (*Bos taurus taurus*) [100]	Montanide ISA 50V (Seppic, France)	Dosage: -100 μg [77,94,99,100] Regime: -Three vaccinations at weeks 1, 3, and 7 [71] -Three vaccinations at weeks 0, 4, and 7 [88,96,98]	1. *R. annulatus* [94] 2A. *R. annulatus* [77] 2B. *R. microplus* [77] 3A. *H. a. anatolicum* larvae [99] 3B. *H. a. anatolicum* unfed adults [99] 3C. *R. microplus* larvae [99] 4A. *H. dromedarii* [100] 4B. *A. cajennense* [100]	Total efficacy: 1. >99.9% [94] 2A. 99.6% [77] 2B. 85.2% [77] 3A. 26.8% [99] 3B. 25.1% [99] 3C. 44.5% [99] 4B. 0% [100] Reduction in tick parameters: 4A. 89% reduction in number of engorging nymphs; 32% reduction in weight of surviving ticks. [100]	Serum antibody levels: 1. N.i. [100] 2. Bm86 mean IgG OD values 2 weeks after 3rd injection (dilution 1: 1000): ±0.2–0.3. [77] 3. Bm86 mean IgG OD values two weeks after 3rd injection (dilution 1: 500): n.i. [99] 4. Bm86 mean IgG OD values two weeks after 3rd injection (dilution n.i.): n.i. [100]	Patent: [78] Literature: [77,94,99,100]
**Ba86** Bm86 orthologue of *R. annulatus*	Crossbreed (n.i.)	Montanide ISA 50V (Seppic, France)	Dosage: -100 μg Regime: -Three vaccinations at weeks 1, 3, and 7	A. *R. annulatus * B. *R. microplus*	Total efficacy: A. 83% B. 71.5%	Serum antibody levels: Bm86 mean IgG OD values 2 weeks after 3rd injection (dilution 1: 1000): ± 1.22.	Literature: [77]
**Bm86** Mozambique strain	1. Crossbreed (n.i.) [77] 2. Holstein Friesian (*Bos taurus taurus*) [101]	1. Montanide ISA 50V [77] (Seppic, France) 2. Montanide 888 (50%) [101]	Dosage: -100 μg [71,102] Regime: -Three vaccinations at weeks 1, 3, and 7 [71] -Three vaccinations at weeks 0, 4, and 7 [102]	1A. *R. annulatus* [77] 1B. *R. microplus* [77] 2A. *H. scupense* [101] 2B. *H. excavatum* [101]	Total efficacy: 1A. 99.6% [77] 1B. 70.4% [77] 2A. 0% [101] 2B. 0% [101]	Serum antibody levels: 1. Anti-Bm86 mean IgG OD values 2 weeks after 3rd injection (dilution 1: 1000): ±1.18 [77] 2. Anti-Bm86 mean IgG OD values in Bm86-vaccinated cattle (dilution 1:1000): [101] -2 weeks after 2nd injection: 1.2 ± 0.1. -2 weeks after 3rd injection: 1.3 ± 0.1. 2. Anti-Hd86 mean IgG OD values in Bm86-vaccinated cattle (1:1000): [101] -2 weeks after 2nd injection: 1.4 ± 0.2. -2 weeks after 3rd injection: 1.6 ± 0.1.	Literature: [77,101]
**rHaa86** Bm86 orthologue of *H. a. anatolicum*	*Bos taurus* X *B. indicus*	1. Saponin in mineral oil [102] 2. 10% Montanide 888 in mineral oil [99]	Dosage: -100 μg [96] -125 μg [103] Regime: -Three vaccinations at weeks 0, 4, and 7 [96,103]	1. *H. a. anatolicum* unfed adults (1st) and larvae (2nd) [102] 2A. *H. a. anatolicum* larvae [99] 2B. *H. a. anatolicum* unfed adults [99] 2C. *R. microplus* larvae [99]	Total efficacy: 1. 61.6% [102] 2A. 68.7% [99] 2B. 45.8% [99] 2C. 36.5% [99]	Serum antibody levels: 1. Anti-rHaa86 mean IgG OD values two weeks after 3rd injection (dilution 1:50): [103] 1.4 ± 0.3. 2. Anti-rHaa86 mean IgG OD values two weeks after 3rd injection (dilution 1: 500): [100] -IgG ± 0.7. -IgG1 ± 0.65. -IgG2 ± 0.3.	Literature: [99,102]
**Ra86** Bm86 ortholog of *Rhipicephalus appendiculatus*	Holstein Friesian (*Bos taurus taurus*)	Montanide ISA 50V (Seppic, France)	Dosage: -50 μg Regime: -Three vaccinations at weeks 0, 5, and 10	*R. appendiculatus*	Reduction in tick parameters: No significant effect on adult engorgement weight, adult mortality, or nymphal mortality. Significant reduction in nymphal moulting and egg viability.	Serum antibody levels: Anti-Ra86 mean IgG OD values 2 weeks after 3rd injection (dilution 1:1000): 0.7 ± 0.1.	Literature: [103]
**Hd86** Bm86 orthologue of *Hyalomma scupense*	Holstein Friesian (*Bos taurus taurus*)	Montanide 888 (50%)	Dosage: -100 μg Regime: -Three vaccinations at weeks 0, 4, and 7	A. *H. scupense* B. *H. excavatum*	Total efficacy: B. 0% Reduction in tick parameters: A. 59.19% reduction in number of engorging nymphs; 0% efficacy against adult tick infestations.	Serum antibody levels: Anti-Hd86 mean IgG OD values in Hd86 vaccinated cattle (dilution 1:1000): -2 weeks after 2nd injection: 1.25 ± 0.5. -2 weeks after 3rd injection: 1.75 ± 0.1. Anti-Bm86 mean IgG OD values in Hd86 vaccinated cattle (dilution 1:1000): -2 weeks after 2nd injection: 1.1 ± 0.1. -2 weeks after 3rd injection: 1.3 ± 0.1.	Literature: [101]
**ATAQ peptide- KLH conjugate** Bm86 homolog	Holstein Friesian (*Bos taurus taurus*)	Montanide ISA 61 VG (Seppic^®^, Paris)	Dosage: -200 μg Regime: -Three vaccinations at weeks 0, 2, and 4	*R. microplus*	Total efficacy: 35%	Serum antibody levels: ATAQ peptide IgG mean OD values 3 weeks after 3rd injection (dilution 1: 300): 0.7 ± 0.3.	Literature: [104]
***Rhipicephalus microplus* Bm86 + H. *anatolicum* Subolesin (SUB) and tropomyosin (TPM)**	*Bos taurus* × *Bos indicus*	Montanide ISA 50 V2 (Seppic, France)	Dosage: -100 μg Regime: -Three vaccinations at weeks 0, 4, and 8	A. *R. microplus* B. *H. anatolicum*	Total efficacy: A. 87.6% B. 87.2% against larvae B. 86.2% against adults	Serum antibody levels: OD values across 6 bovines 2 weeks after 3rd injection for rBm86 (dilution 1:16,000) -IgG: 0.68–0.75 (IgG titre 90 DPI: 1:102,400). -IgG1: 0.68–0.75. -IgG2: 0.25–0.3. OD values across 6 bovines 2 weeks after 3rd injection for rSUB (dilution 1:16,000) -IgG: 0.65–0.85 (IgG titre 75 DPI: 1:64,000). -IgG1: 0.68–0.80. -IgG2: 0.36–0.68. OD values across 6 bovines 2 weeks after 3rd injection for rTPM (dilution 1:8000) -IgG: 0.6–0.72 (IgG titre 90 DPI: 1:51,200). -IgG1: 0.62–0.8. -IgG2: 0.28–0.75.	Literature: [92]

*R.* = *Rhipicephalus. A.* = *Amblyomma. H.* = *Hyalomma*. Total efficacy = 100 [1 − (CRT × CRO × CRF)], where CRT = reduction in the number of adult female ticks; CRO = reduction in oviposition; CRF = reduction in egg fertility as compared to the control group. N.i. = not indicated. IgG = immunoglobulin G. OD = optical density. KLH = Keyhole Limpet Hemocyanin. DPI = days post injection.

However, for South American *R. microplus* strains, a greater level of sequence diversity was observed (up to 8.6%), with an associated efficacy ranging from 58% (Mexican, Mora strain) to 10% (Argentinian, A strain) for the most diverse strains [95]. In 2000, it was shown that the Argentinian *R. microplus* tick populations had a polymorphism in the gene homologous to Bm86, named Bm95, which resulted in variation in the Bm86 sequence between different tick populations [70]. This was suggested as a possible explanation for Gavac^®^’s lower efficacy against tick strains in this region [70]. Consequently, a new recombinant vaccine was produced from the Bm95 variant, called Gavac^®^ Plus, in the hopes that Bm95 would be a more universal antigen to protect cattle against *R. microplus* infestation. However, it was later shown that Gavac^®^ Plus was only more protective than Gavac^®^ in specific geographic regions. For example, Gavac^®^ Plus demonstrated efficacy of 89% and 58% against the Camcord strain (Cuban Bm86-sensitive strain) and the A strain (Argentinian Bm86-resistant, CICV-INTA Castelar strain), respectively, where Gavac^®^ afforded 84% and 0% efficacy, respectively [70]. The Gavac^®^ Plus vaccine also later demonstrated an 81% efficacy against an unspecified Indian field isolate of *R. microplus* [79] (Table 1). A follow-up homologous cattle immunisation and tick challenge study performed in 2023 by Parthasarathi et al. [92] conferred 84.6–88.9% protection from two separate infestations on the same vaccinated animals using Bm86 identified from the IVRI-I Indian *R. microplus* (Table 1).

The Gavac^®^ vaccine was also applied in the late 1990s in the first studies to demonstrate potential cross-species protection (Table 2). A study by Fragoso et al. [94] demonstrated almost 100% protection (>99.9% vaccine efficacy) conferred to vaccinated cattle that were challenged with two separate strains of *Rhipicephalus annulatus* ticks (strains from Mexico and Iran). The high efficacy was hypothesised to be because of high homology between the Bm86 proteins of these two *Rhipicephalus* species, a higher feeding capacity of *R. annulatus* larvae affording a higher uptake of anti-Bm86 antibodies, lower protease content that can break down antibodies in *R. annulatus* saliva, or a combination of these [94]. Additional studies performed in the 2000s evaluated both TickGARD^®^ and Gavac^®^ for cross-species protection (Table 2). Firstly, the high levels of efficacy afforded against *R. annulatus* infestation that were observed in the late 1990s were confirmed in several follow-up studies [77,94,98,99]. Additional cross-species protection was observed against *R. decoloratus* (70%) [98] and *Hyalomma dromedarii* (varying from 32% to 89%) [98,100]. Limited efficacy (25.1% to 50%) was afforded against *H. a. anatolicum* [98,99]. No protection was afforded against *R. appendiculatus* or the *Amblyomma* tick species tested, specifically *A. variegatum* [98] and *A. cajennense* [100].

The observed cross-species protection was proposed to be mediated by immune cross-reactivity between Bm86 and possible orthologous proteins within related tick species. For example, native and recombinant Bd86 proteins (Bm86 orthologue of *R. decoloratus*) were found to strongly react with sera from cattle vaccinated with TickGARD^®^. Subsequent studies identified two conserved linear peptides between Bd86 and Bm86 [105] (Figure 2), which could explain the protection afforded against *R. decoloratus* infestation (Table 2). This is not surprising, as *R. microplus* and *R. decoloratus* are closely related one-host cattle tick species. Further cross-reactivity was demonstrated in rabbits immunised with recombinant Bm86 (*R. microplus*), Ba86 (*R. annulatus*), and Bd86 (*R. decoloratus*) [77]. Although Bm86-based vaccination was found to afford some cross-species protection, evidence suggested that the efficacy of a Bm86-based vaccine may be enhanced when based on the orthologous recombinant Bm86 antigen isolated from the native tick species.

Regarding Bm86 orthologous vaccines, several Bm86 orthologs have been identified to date (Table 2). These include several novel Bm86 orthologues identified by Nijhof et al., such as Rd86-1 and Rd86-2 from *R. appendiculatus*, Ree86 from *R. e. evertsi*, Dr86 in *Dermacentor reticulates*, Hm86 in *H. m. marginatum*, Av86 in *A. variegatum*, Ir86-1 and Ir86-2 in *Ixodes Ricinus*, Is86-1 and Is86-2 in *Ixodes scapularis*, and Os86 in *Ornithodoros savignyi* (the only soft tick sequence available to date) [109,110]. Moreover, Bm86 orthologs were characterised in *H. excavatum* (i.e., He86), *H. dromedarii* (i.e., Hdr86), and *H. m. marginatum* (i.e.,Hm86) with sequence identities ranging between 60 and 66% to *R. microplus* sequences from Australia, Argentina, and Mozambique [111]. In turn, the percent identity between Bm86 orthologs of these *Hyalomma* spp. and an experimental vaccine candidate Hd86 from *Hyalomma scupense* ranged from 87% to 91% [101]. As these are more closely related species, it supports the notion that Hd86 may provide a more appropriate vaccine antigen to target *Hyalomma* tick species than Bm86 vaccines.

The first studies investigating the efficacy of Bm86 orthologous vaccines in bovine vaccination trials were conducted in 2009, including Ba86 (Bm86 orthologue of *R. annulatus*) [77] and rHaa86 (Bm86 orthologue of *H. a. anatolicum*) [102]. Both Ba86 and rHaa86 were found to afford high levels of efficacy in bovine vaccination trials against their native tick species (Table 2), at 83% and 61.6% (against unfed adults), respectively. Ba86 was also found to afford high levels of efficacy against *R. microplus* (71.5%), while rHaa86 provided poor cross-reactivity at an efficacy of 36.5% (Table 2). The latter is consistent, though, with vaccine efficacies conferred by Gavac^®^ of between 25 and 26.8% against *H. a. anatolicum* (Table 2). However, both the Gavac^®^ and TickGARD^®^ antigens still provide better protection against *R. annulatus* infestations than the native Ra86 orthologue (Table 2). Some life-stage-specific protection was identified in that rHaa86 vaccination afforded higher levels of protection against the larval life stage of *H. a. anatolicum* ticks relative to adult ticks (61-68% versus 45.8%, respectively) [99,102] (Table 2). In addition, Hd86, the Bm86 orthologue of *Hyalomma scupense* [101], and Ra86 (the Bm86 orthologue of *R. appendiculatus*) [103], also only afforded protection against the immature life stages of the tick (Table 2).

Vaccination trials were also conducted to test the efficacy of recombinant Ir86 antigens (i.e., rIr86-1 and rIr86-2) in rabbits infested with *I. ricinus* ticks [112]. Though high IgG titres against the Ir86 antigens were produced (1:10^5^ and 1:10^6^ for rIr86-1 and rIr86-2, respectively), no significant effect on tick biological parameters was observed [112]. This shows that the Ir-86 antigens are likely not a viable option for a novel vaccine to control *I. ricinus* infestations. This highlights that even though an immune response can be raised against a Bm86-based antigen, this does not necessarily correlate to the antigen providing protection against tick infestation. Despite the reported levels of protection conferred within and between tick species (Table 2), no species-specific vaccines or combinatorial vaccines containing the most prevalent (and protective) native Bm86 orthologues have been investigated for commercialization to date. Only one recent study by Parthasarathi et al. [92] demonstrated cross-species protection between Indian IVRI-II strains of *R. microplus* (84.6% and 88.9% vaccine efficacy) and *H. anatolicum* (86.2% efficacy against adult infestations) by using a multi-antigen vaccine containing Bm86 derived from the *R. microplus* strain, as well as Subolesin and tropomyosin isolated from the *H. anatolicum* strain, respectively. The studies described above highlight the fact that other factors besides sequence homology might be playing a role in Bm86-based vaccine efficacy. One such factor is protein expression levels in the various tick tissues, including over the ticks’ life cycle, that could contribute to variation in vaccine efficacy. A study by Nijhof et al. [109] demonstrated that the amino acid sequences of two *R. appendiculatus* Bm86 orthologues, Ra86-1 and Ra86-2, have sequence identities of 72.9% and 73.8% with the TickGARD^®^ antigen from *R.* (*B.*) *australis*, as well as 73.7% and 74.6% with the Bm86 sequence from a Mozambican strain of *R. microplus*, respectively. When evaluating the expression profiles, Bm86 protein expression was reported to be more continuous throughout the life cycle of the one-host tick *R. microplus*, as opposed to the large variation observed in the expression profile of Ra86 between the different life stages of the three-host tick *R. appendiculatus* [109]. Expression levels of Ra86 were also found to be significantly higher in *R. appendiculatus* than the expression of Bm86 in *R. microplus* for eggs, unfed larvae, and unfed nymphs. Although in both tick species Bm86 expression decreased significantly during feeding and moulting (to a lesser extent in *R. microplus*), the expression levels of Bm86 were similar in the adult ticks of both species. An additional study confirmed that Bm86 is indeed differentially expressed at different tick life stages [113], even though the expression levels may appear to be more continuous than for other tick species [109]. This study also showed that although the highest expression of Bm86 occurs in the female midgut, Bm86 is also expressed in the ovary tissues [113]. Although this alone is not sufficient to explain the differences in vaccine efficacy observed.

Lastly, in 2010, a novel Bm86 homologue called ATAQ (or BmATAQ) was identified, which was shown to have 40% sequence homology to the TickGARD^®^ Bm86 antigen of *R.* (*B.*) *australis* [110]. ATAQ was initially identified in *Amblyomma variegatum* ticks, with orthologs subsequently identified in *Rhipicephalus annulatus*, *R. decoloratus*, *R. microplus*, *R. e. evertsi*, *R. appendiculatus*, *Hyalomma marginatum*, *Dermacentor reticulatus*, *D. variabilis, and Haemaphysalis elliptica*. ATAQ was shown to also have multiple epidermal growth factor (EGF)-like domains, just like Bm86. It was demonstrated that ATAQ is expressed in both tick midguts and Malpighian tubules, whereas expression of Bm86 orthologues is restricted to the tick midgut. The expression of ATAQ was also found to be more continuous across the life stages of both *R. microplus* and *R. appendiculatus* ticks than Bm86 [110]. A recent vaccination and protection study using a short peptide designed from BmATAQ, with or without conjugation to Keyhole Limpet Hemocyanin (KLH), conferred cross-protection in vaccinated rabbits (29% and 47% efficacy, respectively) against *R. sanguineus* [104]. A vaccine efficacy of ~35% was obtained in cattle vaccination trials against *R. microplus* infestations [104]. An ATAQ orthologue, rHlATAQ, was also tested in rabbit vaccination trials infested with *Haemaphysalis longicornis* with little effect [114]. It was suggested that this antigen could be better utilised as a component of a multivalent vaccine to improve the efficacy of other vaccine antigens [104]. However, the efficacy of this antigen in the field will likely suffer from the same strain variations in antigenic sequence and vaccine efficacy challenges faced with Bm86-based vaccines [115].

## 5. The Next Generation of Biotechnology and Tick Vaccines (2000–Present): New Advances in Tick Vaccine Development and Design

In the 2000s, several new biotechnologies became available that impacted tick vaccine development (summarised in Figure 1), such as expression library immunisation (ELI) [116], DNA microarrays [117], RNA interference (RNAi) [118], and next-generation sequencing (NGS) technologies [119]. Following the introduction of NGS, the sequencing of the first tick genomes was initiated, with the *I. scapularis* [120] and *R. microplus* [121] genome projects starting in 2005 and 2006, respectively. This was followed by the development of a genome assembly draft for *R. microplus* in 2017 [122] and assembled genomes for *R. microplus* and *R. annulatus* in 2021 [123]. Only this year (2023), the first high-quality tick genome was published for *I. scapularis* [124]. RNA sequencing was first reported in studies conducted in 2006 [125,126], and it was shortly applied to ticks in 2007 [127]. DNA microarrays, on the other hand, took just over a decade to be applied to ticks [128]. All these technologies have ultimately allowed for the downstream implementation of efficient reverse vaccinology approaches to rapidly screen for and identify potential protective tick antigens and/or to better elucidate the functionality of existing antigens [129]. However, no new, effective commercial vaccines have yet been registered as a result of reverse vaccinology studies.

The first RNAi study in ticks was conducted in 2002, where the applicability of the double-stranded RNA (dsRNA) technique in *Amblyomma americanum* female ticks was tested by silencing the expression of histamine binding proteins (HBPs), a salivary gland protein involved in tick feeding [130]. In 2003, the first study to identify protective antigens for control of *Ixodes scapularis* using ELI combined with analysis of expressed sequence tags (EST) was conducted in a mouse model of tick infestations [131]. It was in this study that Subolesin was identified as a protective antigen, then known as 4D8. To further simplify the screening methods for the identification of tick protective antigens, RNAi was used two years later to screen the same pools of *Ixodes scapularis* cDNAs used previously in the ELI experiments [132]. The RNAi results reflected what was obtained by the ELI studies, where Subolesin (4D8) significantly affected tick biological parameters [132].

In 2007, the first gene silencing of Bm86 was conducted along with two other *R. microplus* protective antigens, Bm91 and Subolesin [133]. Significant changes in tick fecundity were not observed for either Bm86- or Bm91-silenced *R. microplus* females. This led the authors to propose that the effects associated with vaccination are not due to a loss of protein function alone. For example, vaccination with Bm86 is associated with gut damage [45], which may occur even in the presence of limited amounts of Bm86 protein. Decreased tick and egg weights, as well as increased mortality rates, were observed for Subolesin-silenced ticks [132,134] (Table 1). These results are not that surprising when considering the long half-life of membrane proteins such as Bm86 and the fact that mRNA depletion may not impact protein levels to the extent that a phenotype would arise. In contrast, silencing of transcription factors and proteins associated with the regulation of gene expression, such as Subolesin, will have a cascading effect and may easily result in a phenotype affecting tick biology.

## 6. Advances in Bm86 Epitope Mapping—Entering the Era of Peptide Vaccines

In the early 1990s, attempts were made to identify the regions in the Bm86 antigen that contribute to immunoprotection [49]. In these early studies, five protein constructs were produced in *E. coli* for the *R. australis* Bm86 antigen, consisting of one full-length protein (i.e., amino acid position 31-629) and three C-terminal truncations (i.e., amino acid positions 31-406; 31-223; and 31-88) containing an N-terminal β galactosidase fusion tag, as well as one N-terminal truncation (i.e., amino acid position 351-576) lacking the fusion tag [49]. Cattle vaccination and tick infestation studies showed that two non-overlapping regions, i.e., amino acid positions 31-223 and 351-576, contributed significantly to protection, with reductions in reproductive efficacy of 53% and 65%, respectively [49]. Interestingly, from the data, the N-terminal fusion tag seems to contribute significantly to the antigenicity of the C-terminal truncations (amino acids 31-223) (Figure 2). This is the first illustration of how conjugation of a less protective antigen to a synthetic tag can increase its overall antigenicity. Moreover, it can be hypothesised that the primary protective epitopes for the Bm86 antigen may occur in the untagged C-terminal region (amino acids 351–576) [49].

In 2002, the first Bm86-based peptide vaccines were developed and patented [84], although none have yet been commercialised (Figure 2). Bm86-based epitope vaccines can potentially be advantageous over recombinant protein vaccines as they involve reduced production costs and possibly increased vaccine stability. The synthetic Bm86 epitope constructs SBm4912^®^ and SBm7426^®^ were predicted from the Bm86 sequence of the Yeerongpilly strain of *R.* (*B.*) *australis* based on physicochemical properties of the amino acids in the primary sequence (e.g., hydrophilic, and hydrophobic properties), as well as propensities for secondary structures (e.g., α-helix- and β-sheet-forming potentials). These peptide constructs were found to afford higher efficacy in controlled bovine vaccination trials than some of the Bm86 vaccines using the full recombinant antigen (Table 1). The results of the initial vaccine trials showed that the order of the peptides in the epitope vaccine constructs played a role in the efficacy that was afforded. Bm4912^®^ and SBm7426^®^ comprise the same peptides in different orders, yet SBm7426^®^ afforded higher levels of protection against *R. microplus* infestation. It is likely that the last peptide in the Bm86 sequence has the most impact on vaccine efficacy, as moving this peptide to the middle of the construct sequence increased vaccine efficacy (peptides 4822, 4823, and 4824 in SBm4912^®^ and peptides 4822, 4824, and 4823 in SBm7426^®^) (Figure 2). Moreover, peptide 4822 occurs within the C-terminal region previously identified by Tellam et al. [49]. A sequence diversity study conducted by Peconick et al. [135] showed that despite some differences in nucleic acid sequences, the predicted protein epitope sequences identified by Patarroyo et al. [84] were highly conserved among South American *R. microplus* strains, as well as for most international strains, especially peptide 4824 (Figure 2). The latter peptide’s placement within the SBm7426^®^ construct may provide better processing and presentation of this protective epitope, leading to the observed vaccine efficacy of 72.4% and 81.05% (Table 1, Figure 2).

In 2009, SBm7426^®^ was evaluated further towards understanding the underlying immune responses activated by vaccination beyond the simple measurement of serum antibody levels [72] (Table 1). Bovine lymph node tissue histology was conducted, which showed an increase in the germinal centres of B-cell follicles, hyperplasia of the medullary cord, and the presence of antigens in antigen-presenting cells (APCs). This study provided limited evidence of T-cell activation, using the hyperplasia of T-dependent areas in lymph nodes and the presence of PAP-positive APC cells as proxies for T-lymphocyte interactions. An IgG1-dominated anamnestic response was identified that could point towards a T-helper-2-mediated pro-inflammatory response. Additional limited phenotyping of circulating peripheral blood mononuclear cells (PBMCs) was conducted using only two markers for flow cytometry, CD21 (a B-cell marker) and CC101 (a WC1+ γδ T cell marker). A significant increase in CD21 was reported at 15 and 70 days after immunisation, indicating B-cell activation, which is consistent with the increase in antigen-specific antibodies. This is also consistent with the antigen-specific proliferation observed in the lymph nodes during germinal centre development and the presence of antigen-specific antibodies [72]. Variation was observed in CC101-positive cells, where only a slight, non-significant increase was observed at 35 days (5 days after the 2nd immunisation).

Another companion study conducted later in 2020 included cytokine gene expression profiling using total RNA isolated from cultured PBMCs [87]. Five markers were included for gene expression analysis of cytokines via qPCR, including Il-4, IFN-y, IL-10, IL-12, and TNF-α. Of the cytokines analysed, only IL-4 was significantly upregulated, which demonstrated a polarisation towards a type 2 response. IL-4 is produced by T-cells to co-stimulate B-cells to differentiate in a class switching to high-affinity IgG1 antibodies. This is consistent with the increase in CD21+ B cells and IgG1 class observed previously when vaccinating with synthetic peptide [72]. The main evidence produced by both studies for immune mechanisms induced by Bm86-based vaccination is the activation of B cells and the production of IgG1 antibodies, which is consistent with what has been published previously.

Some limitations regarding both these studies include: (a) a lack of significant results that limit the conclusions that could be drawn regarding the involvement of γδ T cells or other cytokines aside from Il-4 in host protective responses; (b) the PBMCs used in the 2009 study were not stimulated with the SBm7426 peptide and/or Bm86 antigen to investigate the antigen-specific recall responses; (c) antibody titres were not determined in both studies; and (d) the impact that tick infestation would have on host responses was not considered in the 2009 study. Although the antigen-specific antibody responses in both studies were significant, in the absence of a proper titre determination, serum antibody dilutions used in serological assays (1:100 to 1:400) indicate a low level of host serum responses. Infestation certainly decreases the observed antibody levels, which is evident in the lower dilution used in serological assays (i.e., 1:100) performed when vaccinated cattle were infested with ticks in the 2020 study versus vaccinated but uninfected cattle in the 2009 study [72,87]. Aside from the host level of antibodies decreasing due to tick feeding, the antibody response is known to be a target for tick modulation/evasion mechanisms, such as immunoglobulin binding proteins (IGBPs) that capture, transport, and inject antibodies back into the host [136]. This calls into question whether this peptide or any other Bm86-based vaccine can properly confer protection on cattle hosts with heavy tick burdens in the field.

To further improve antigenic peptide selection for vaccine design, a study produced anti-*R. microplus* hyperimmune sera by injecting chickens with tick proteins isolated from adult female and larval tick tissues [108]. These hyperimmune sera were then used in the bio-selection of random peptides produced from six distinct peptide phage-display libraries, and two epitopes (i.e., TPDKS and LHXXL) related to Bm86 were discovered for *R. microplus* (Figure 2). The latter marked the first and only use of phage-display for antigen discovery in ticks since the discovery of phage-display in 1985 [137] (Figure 1).

In more recent years, in silico epitope prediction tools have been extensively utilised to evaluate which regions of Bm86 could be recognised by B- and T-cells and the major histocompatibility complexes (MHCs), which could potentially be candidates for peptide-based vaccines against *R. microplus* [106,107]. A 2021 study recently utilised total RNA to amplify full-length Bm86 sequences from Indian isolates of *R. microplus* to map the variability of the Bm86 protein in Indian strains and to predict B-cell epitopes for vaccine development [107]. This study identified nine B-cell epitopes, of which at least three epitopes showed a high degree of variability between various geographical strains of *R. microplus* (incl. Australia, Thailand, USA, and China) (Figure 2). Based on the high degree of polymorphisms found in Bm86 gene fragments amplified from *R. microplus* ticks collected from nine states of India, the authors concluded that a country-specific multi-epitope Bm86 vaccine would likely afford higher efficacy than a single full-length antigen. Although good protection against an Indian field isolate of *R. microplus* was shown by Kumar et al. [79], where the recombinant Bm95 antigen from the Argentinean strain A of *R. microplus* was used (i.e., vaccine efficacy of ~81%). A recent study by Parthasarathi et al. [92] using a multi-antigen vaccine including a full-length Bm86 produced from the Indian IVRI-I strain did, however, produce superior protection of more than 88%. In turn, Blecha et al. [106] conducted in silico prediction of linear T- and B-cell epitopes using both bovine MHC I (via BoLA loci) and mouse MHC I and II (via the H-2 locus) alleles for prediction of antigenic peptide sequences from various global *R. microplus* geographical strains. Additional criteria were also considered, such as the presence of signal peptides, membrane topology, glycophosphatidylinositol anchor regions, and intrinsically disordered regions. Finally, three major, relatively conserved antigenic regions were identified. From these results, the authors proposed that a universally protective Bm86 peptide-based vaccine could be possible. However, this remains to be demonstrated.

To conclude, prediction of immunogenic epitopes should be based on the objective of selecting peptide sequences that are capable of binding to all bovine MHC allelic variants, triggering a humoral immune response, and stimulating immune memory [106]. To achieve this, it is vital to understand the diversity of bovine MHC alleles in a population and the parameters that determine which peptides can be bound and presented to B- and T-cells to accurately predict the potential of antigens to elicit an immune response [138]. Whether the immune response elicited would confer protection against infestation in vivo remains difficult to predict, and final validation in animal vaccination and infestation studies remains paramount. Previous studies were limited to the prediction of immunogenic linear B- or T-cell epitopes for Bm86 based predominantly on murine or human MHC alleles [106,107]. Recently, more extensive databases and tools for investigating bovine MHCI and MHCII alleles have become available [139], allowing for more accurate in silico prediction to be conducted for bovine MHCI and MHCII epitopes. This ultimately paves the way for more accurate Bm86 epitope mapping to be conducted in the future.

## 7. Current Bm86-Based Commercial Vaccines and the Next Generation of Bm86-Based Cocktail Vaccines

Although several new vaccine candidate antigens have been identified over the years, Bm86-based vaccines remain the most cost-effective against *R. microplus* infestations under field conditions to date [13,129,140]. A recent study investigated the effectiveness of applying the Gavac^®^ vaccine in a national *R. microplus* tick control program in Venezuela over two years [141]. This study included more than 1.9 million bovines across 18 states and 38,835 cattle farms and resulted in a reported reduction in the use of acaricides by 83.7% (more than 260 tons of chemical acaricides). Furthermore, an 81.5% reduction in the costs of acaricide treatments was reported. This ultimately gives strong support for the use of Bm86-based vaccines in tick control.

Both TickGARD^®^ and TickGARD^®^ Plus were discontinued by 2010 [18]. Firstly, this was due to vaccination requiring 3 to 4 boosters per year, hampering the adoption of the TickGARD^®^ vaccine by the Australian beef industry, where cattle were mustered only once per year [47]. Secondly, variation in vaccine efficacy in the field, as well as limited cross-protection to other tick species, further hampered the expansion of TickGARD^®^ into other markets [18]. Initial evaluations of Gavac^®^ in the field suggested that after three initial immunisations, the duration of the immunity induced is around 5-6 months and that a single yearly booster is required to maintain immunity [61]. A later study with Gavac^®^ Plus indicated the possibility of improving the vaccination regime, where two initial vaccinations followed by a yearly booster were sufficient to confer protection against *R. microplus* infestations under production conditions [80]. This contributes to the lower costs of implementing vaccination in animal production processes. Both Gavac^®^ and Gavac^®^ Plus vaccines are still commercially available from Herber Biotec S.A., Cuba. A new Bm86-based vaccine, Bovimune-ixovac^®^, was also developed and is available from Lapisa^®^ (Michoacán, Mexico) [106]. One other commercial *R. microplus* tick vaccine (i.e., Go-Tick^®^ or Tick-Vac^®^), based on fractionated larval extracts, is currently available in Latin America from Limor de Colombia^®^ (Bogotá, Colombia) [34].

To further enhance the protection of Bm86-based tick vaccines, multi-antigen or “cocktail” vaccines combining two or more protective antigens have been explored [140,142] (Table 1). Ticks have very large genomes that are highly repetitive, which likely results in the expression of a plethora of proteins depending on their environment and their life stage [122,123,143]. For this reason, it is hypothesised that multi-antigen vaccines could enhance the level of protection afforded against ticks [47]. To date, there are a few multi-antigen vaccines, including Bm86, that have been tested in bovine vaccination trials against *R. microplus* infestation, achieving over 50% efficacy (Table 1), pointing towards possible synergistic effects [89,92,142].

However, multi-antigen vaccines do not always have a synergistic effect, and decreased efficacy has also been reported. For example, Bm86 and a synthetic peptide derivative of the Subolesin antigen (pSUB) were recently tested in bovine vaccination trials [91] (Table 1). The antigens alone were found to afford higher protection (pSUB alone: 67%; Bm86 alone: 56%) than when the antigens were combined (49%). A similar study by Parizi et al. [144] investigated a multi-antigen vaccine composed of rGST-Hl from H. *longicornis* ticks, vitellin-degrading cysteine endopeptidase (VTDCE), and *Boophilus* yolk pro-cathepsin (BYC) from *R. microplus* ticks against *R. microplus* tick infestation in cattle, where no enhanced protection was observed and a reduction in antibodies against each of the antigens was noted. This could be due to several factors such as antigenic competition, antigen concentration, antigen-adjuvant interaction, or bovine genetics [140].

Chimera-based multi-antigen vaccines (hybrid vaccines where two or more antigenic fragments are fused together) have also been investigated, although studies are limited and none have been commercialised to date [140]. For example, a chimeric vaccine protein consisting of *R. appendiculatus* Bm95 orthologous antigenic peptides fused to the complete SUB protein, which in turn was fused to an *A. marginale* MSP1a protein, was surface expressed on *E. coli* [88]. Consequent vaccination trials using the bacterial membrane fraction containing recombinant chimeric protein resulted in a 60% reduction in *R. microplus* infestations and 91% total efficacy considering the reduction in viable progeny [88]. However, this study was only a preliminary cattle trial, and further research has not been published.

Both tick- and/or host-based biological factors, as well as the vaccine antigen and/or formulation used, can contribute to variations in observed vaccine efficacy. From Table 1 and Table 2, it is evident that there are challenges with Bm86-based vaccines that have not been sufficiently addressed in the past decades of research and development.

## 8. Bm86-Based Vaccines: What Are the Current Gaps and Future Research Needed?

### 8.1. Antigen Stability and Adjuvants

Antigen stability (their ability to retain their structural integrity and immunogenicity during the storage and administration of vaccines) is well documented in the scientific literature [145,146,147,148]. A stable antigen ensures that the vaccine maintains its potency and effectiveness throughout its shelf life. Factors such as temperature, pH, and exposure to light can influence antigen stability and, as such, proper storage conditions (e.g., refrigeration or freezing) are essential to preserve the antigen’s integrity. If an antigen becomes unstable, it may lose its ability to induce a robust immune response, leading to reduced vaccine efficacy. As such, it is surprising that these basic experiments have never been published before for the Bm86 antigen. Only one study was conducted by Vargas-Hernández et al. [149] investigated the stability of the Gavac^®^ vaccine when subjected to heat stress, as this is important for the downstream vaccine shelf life and the need for a cold chain. It was reported that even after 15 days of heat stress at 37 °C, Gavac^®^ was still able to elicit antibody titres (1:5435) 56 days after first immunisation (injecting at day 0 and day 28). The response was significantly lower than that observed for intact vaccine (1:28 353). However, the heat-stressed vaccine was still able to significantly affect the reproductive parameters of ticks to a similar degree as the intact vaccine control for all parameters tested [149].

Improving the understanding of the potential impact of Bm86 instability on variable protection observed during field trials remains a paramount objective. For the large-scale production of recombinant vaccine antigens, it is often challenging to ensure that recombinant proteins maintain immunological activity that is comparable to the native parasite protein [34]. For instance, considering that the predominant immune response to the Bm86 antigen is humoral, i.e., antibody production, the stabilisation of the tertiary structure of the Bm86 antigen is essential to ensure antigen recognition by B-cells via the B-cell binding receptor (BCR), which will recognise and bind to the antigen and initiate subsequent antibody production [150]. Using structural vaccinology approaches, it could be possible to adapt the Bm86 antigen to simplify the protein for large-scale production while maintaining its antigenicity and increasing its stability [151,152,153]. Additionally, selection of the best protein production platform should be based on the antigen with the best techno-economic profile based on safety, ease of production, cost of processing, scalability etc. For example, studies conducted in the 1990s with the TickGARD^®^ antigen showed marked differences in the reduction in tick reproductive capacity when using native Bm86 protein (between 91 and 93%) and proteins produced recombinantly in bacteria (68–89%), yeast (70–75%), as well as insect cells via baculovirus-mediated expression (91%) [49,58]. Subsequent studies proceeded with production of Bm86 in a yeast system for commercialization as it affords the necessary post-translational protein modifications and is generally accepted as being a safe vaccine production platform [154,155].

Adjuvants play a crucial role in veterinary vaccines by enhancing the immune response to antigens [156,157]. Adjuvants serve as immune potentiators, stimulating and directing the immune system’s response to the antigen. They work by activating immune cells, such as antigen-presenting cells like macrophages and dendritic cells and promoting the release of cytokines and chemokines. This activation leads to a stronger and more durable immune response, including the production of antibodies and the activation of cellular immunity. Adjuvants also help with antigen persistence by improving the antigen’s retention at the site of injection and promoting a prolonged immune response. They can enhance vaccine efficacy by allowing the use of lower antigen doses, reducing the number of vaccine doses required, and increasing the duration of protection. As such, the evaluation of adjuvant effects in the formulation of Bm86 vaccines remains a large gap in the field. For tick vaccines, oil-based emulsions have predominantly been used as adjuvants in successful vaccine formulations, such as Freund’s Complete Adjuvant and Montanide™, as well as some saponin adjuvants (Table 1 and Table 2). Both commercialised Bm86-based vaccines, Gavac^®^ and TickGARD^®^, are formulated with Montanide™ (water-in-oil adjuvant). Montanide™ functions as a depot delivery system at the site of injection, enabling slow antigen release and stimulation of antibody producing plasma cells [158,159], although the type of immunity induced (i.e., Th1 versus Th2) is uncertain and likely dependent on the antigen [160,161].

Adjuvants such as Montanide™ and saponin can cause systemic side effects such as mild to severe local inflammation and the formation of granulomas at the site of injection [162,163]. The adverse inflammatory reaction can trap vaccine antigens at the injection site, limiting the host immune response [164]. In addition, the inflammation and granulomas can cause tissue damage that results in unwanted meat reduction and damage to hides in cattle, which reduces the value of cattle products and incurs losses to farmers [165,166]. For Montanide™, it is also difficult to achieve proper emulsion of the adjuvant and the target antigen in the vaccine formulation [163]. This ultimately evidences the limited suitability of Montanide™ as a tick vaccine adjuvant and highlights the gap and the need for testing of more suitable adjuvants.

### 8.2. Understanding the Immune Responses That Confer Protection across Genetically Diverse Bovine Breeds: The Need to Identity Shared Elements for Targeted Intervention

A thorough understanding of the host immune system is essential for designing effective vaccines, and by studying the intricacies of this system, researchers can identify the key components and mechanisms involved in mounting a protective immune response. For Bm86, limited immune markers were investigated across all studies. The antibody response was predominantly investigated via antibody titres. Additionally, studies mainly reported on the total IgG antibody response to vaccination, where few studies reported an IgG1 isotype, thus providing limited evidence of a type 2 immune response. There is also a large variation in the antibody titres that were reported across all the studies, and comparability between studies is limited as different serum dilutions were used and, in several instances, serum dilutions were not indicated. Only the two studies surrounding the evaluation of the Bm86-based peptide vaccines SBm4912^®^ and SBm7426^®^ investigated additional immune markers/mechanisms [72,87] (Table 1), but these are questionable due to the lack of PBMC stimulation and the very low antibody titres reported.

Additionally, understanding the duration of the immune response and memory formation assists in developing vaccines that confer long-lasting protection. Since tick infestation and feeding does not naturally boost host immunity in Bm86-vaccinated animals booster immunisation is required. In this regard, long-term studies on cattle vaccinated with Bm86-based vaccines are unavailable. As stated before, knowledge of the host immune system allows for the identification of potential adjuvants or delivery systems that can enhance the immune response to the vaccine. This rational design of Bm86-based vaccines and tick vaccines in general remains unexplored (Table 1).

Insights into the immune response have significantly influenced vaccine design in animals. Some examples include (a) Porcine Circovirus: research on the immune response to Porcine Circovirus revealed the crucial role of cell-mediated immunity in controlling the infection [167,168,169]. This understanding has guided the design of vaccines that stimulate both humoral and cellular immune responses, contributing to improved protection against the virus in pigs. (b) Bovine Respiratory Syncytial Virus: investigations into the immune response to Bovine Respiratory Syncytial Virus have identified the importance of cell-mediated immunity, particularly cytotoxic T lymphocytes, in controlling the infection [170,171]. This knowledge has informed the development of vaccines that elicit strong cellular immune responses, reducing viral replication and improving protection in cattle, and (c) insights into cellular immunity against *Ehrlichia*, the causative agent for East Coast Fever [172,173,174].

Considering the diverse genetic composition of different bovine breeds, understanding the host immune response is only one aspect of a multifaceted system. Despite the evidence that the host response against ticks is a multi-factorial trait that involves multiple host-related factors, understanding of the biological mechanisms has only been described in a few publications [31,175,176,177,178,179]. As such, this remains a large gap in the rational design of tick vaccines. Data pertaining to the bovine breeds used during field vaccination trials would have greatly contributed to understanding the successes and failures of Bm86 vaccines.

In the new era of vaccine design, in silico tools hold a lot of promise. However, we are facing the obstacle of a lack of immune-informatics tools for non-model organisms such as bovines. It is known that there is large gene diversity in the major histocompatibility complex (MHC) genes of cattle [139], which can play a role in tick resistance or susceptibility [180] as well as influence the immune response against vaccines [106]. The MHC locus contains genes that play key roles in initiating and regulating immune responses, including polymorphic MHCI and MHCII genes, which present peptide antigens to CD8+ and CD4+ T-cells, respectively [139]. The diversity of MHC alleles in a population will thus likely play a role in determining which antigenic peptides can be bound and presented to elicit T cell responses [138]. There is genetic variation both between and within the bovine MHCI and MHCII molecules. Recombination events can occur at positions between the MHCI and MHCII loci, which will generate a range of different MHCI-MHCII haplotypes [139]. Within the bovine MHCI gene, six gene loci have been proposed [181,182]. The number of MHCI genes expressed varies between haplotypes with varying permutations, which contributes to the diversification of the MHCI haplotype repertoire along with possible MHCI allele mutations [139].

Within the bovine MHCII gene locus, there are two categories of conventional MHCII molecules: the bovine leukocyte antigen class II (BoLA) DQ locus (BoLA-DQ) and DR locus (BoLA-DR). The BoLA-DQ includes both DQA and the DQB gene loci, which are highly polymorphic and can contain up to five genes each [183,184]. There is variability between haplotypes in the number of DQA and DQB alleles present, with approximately half of the haplotypes expressing single DQA/DQB genes and half expressing two [185]. BoLA-DR includes both DRA and DRB loci, where, similarly to other species DRA, is essentially monogenic [186]. DRB contains three loci, of which only one locus (DRB3) is functionally relevant, as DRB1 is a pseudogene and DRB2 is expressed at very low levels or not at all (it is often not functionally transcribed) [187]. The variability of the DRB locus can thus be characterised via the alleles present at the DRB3 gene locus.

Due to the large diversity of the bovine MHCI, DQA, and DQB genes, there is a lack of large-scale studies of their allelic repertoire, whereas the DRB3 gene has been most extensively utilised in studying bovine MHCII diversity across the globe due to its polygenic nature [188,189,190]. Recently, bovine DRB3 gene analysis was incorporated into a high-throughput sequencing platform for bovine MHC genotyping, which provides an integrated system where bovine MHCI and MHCII repertoires can be examined in parallel [139]. Further expansion of the platform to incorporate analysis of the BoLA-DQ genes will enable high-throughput examination of the entire classical MHC repertoires of cattle populations [139]. Combined with immunopeptidomics, this may be a useful platform for the development of vaccines that aim to induce protective T-cell responses in cattle [139].

### 8.3. Impact of Field Conditions: Co-Infestation and Co-Infection

Co-infections (the simultaneous presence of multiple infectious agents in an organism) can have a significant impact on the immune response and the effectiveness of vaccines. Co-infections can result in immunomodulation, where the immune response to one pathogen may be altered by the presence of another. This modulation can manifest in various ways, such as the suppression or enhancement of immune responses, changes in the production of cytokines, or alterations in the activation and function of immune cells. Consequently, co-infections can pose challenges for the development and efficacy of vaccines. Additionally, co-infections may create immune memory that differs from that generated by single infections, impacting subsequent responses to future infections or vaccinations. Understanding the interplay between co-infections and the immune system is crucial for optimising vaccine strategies and developing effective interventions in animal health.

When analysing bovine health, there is a growing body of evidence for the high burden of internal parasites [188,189,190]. When we consider the impact of co-infections involving parasites on vaccine responses in animals, a new challenge arises in transitioning tick vaccines from controlled laboratory settings to real-world field conditions. Several publications on the effects of co-infection with gastrointestinal parasites on the efficacy of vaccines in animals are available [191,192,193,194,195]. Along the same lines, co-infestation with different tick species and their associated tick-borne pathogens [196] may greatly affect the host’s immune response to tick infestation, transmission of tick-borne pathogens, and vaccination. For instance, the interactions between the tick vector and pathogen infections within a host can have synergistic or antagonistic effects, resulting in diverse effects on host susceptibility, infection duration, transmission profile, and clinical manifestations [197,198,199]. It is thus plausible to hypothesise that co-infestation of *R. microplus* and the infectious pathogens it transmits during feeding (i.e., *Babesia* and *Anaplasma* spp.) can potentially influence the bovine host immune system and play a role in affecting vaccine efficacy. Thorough data collection during future vaccination trials, encompassing breed, infection status, weight, and hematocrit, is essential. Instead of solely focusing on tick load and IgG titres, leveraging advanced technologies like next-generation sequencing and high-throughput diagnostics can enable us to achieve this approach and identify vaccine-efficiency-influencing factors. Implementing these methods will prove invaluable in successfully translating tick vaccines into real-world field conditions.

## 9. Conclusions

After 37 years of research on the Bm86 antigen, the pertinent question that we are faced with is: “Has Bm86-based vaccination achieved its full potential, and is it time to move on to other potentially better vaccine candidates?”

The Bm86 antigen is the best model tick antigen and remains relevant even today. But it is also still an enigmatic black box. This ‘gold standard’ for tick vaccines was discovered at a time when the technological resources of the day were limited. With all the current technologies at our disposal, rational targeted research is needed to, for example, fully characterise: the Bm86 antigen, its antigen stability, and the influence this has on antigen presentation to bovine MHC I and II; the optimal vaccine type (i.e., subunit protein, DNA, or RNA); the formulation (i.e., what adjuvant and what immuno-stimulatory conjugates and/or additives); and the host contributing factors (i.e., immune correlates of protection, host genetics, and the role of co-infections). With all these gaps, the need for rational and systematic research endeavours to gain quality scientific data useful to guide us into the future is paramount. There is still so much to explore and learn from this antigen, and we look forward with anticipation to what the next decade will reveal and contribute to tick and tick-borne disease control.

## Figures and Tables

**Figure 1 pathogens-12-01071-f001:**
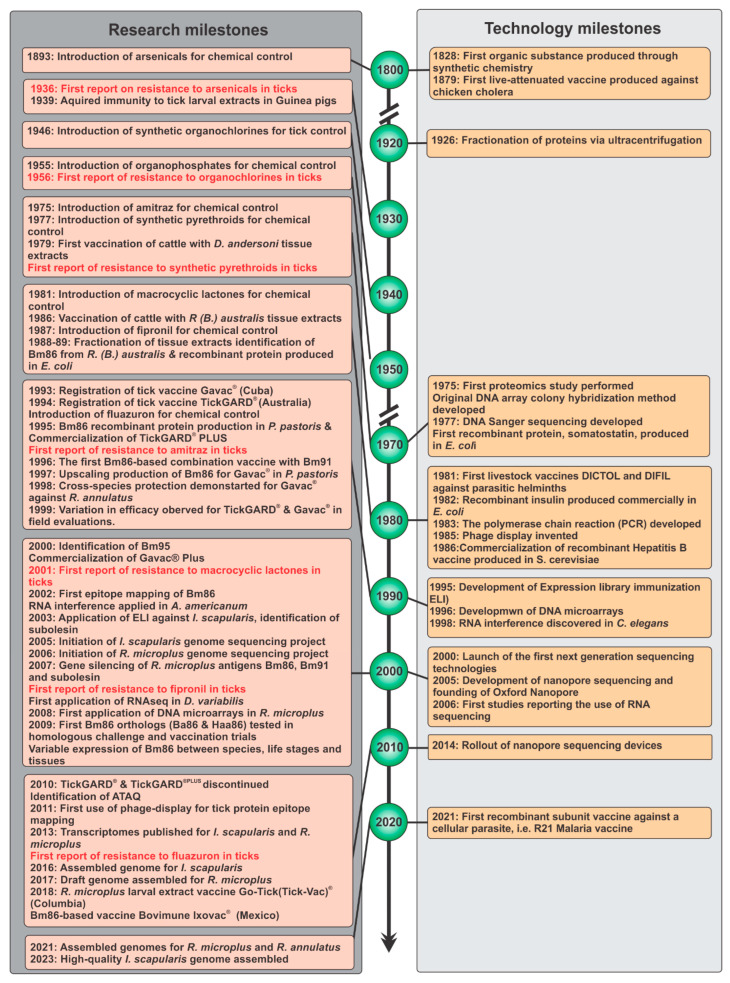
Discovery and development timeline for advancement of tick vaccine development according to date of first research publication. Highlighted are some research and technology milestones from the 1800s to the present. Acaricide resistance development of ticks to different acaricide classes are indicted in red. Additional information is supplied in Appendix A.

**Figure 2 pathogens-12-01071-f002:**
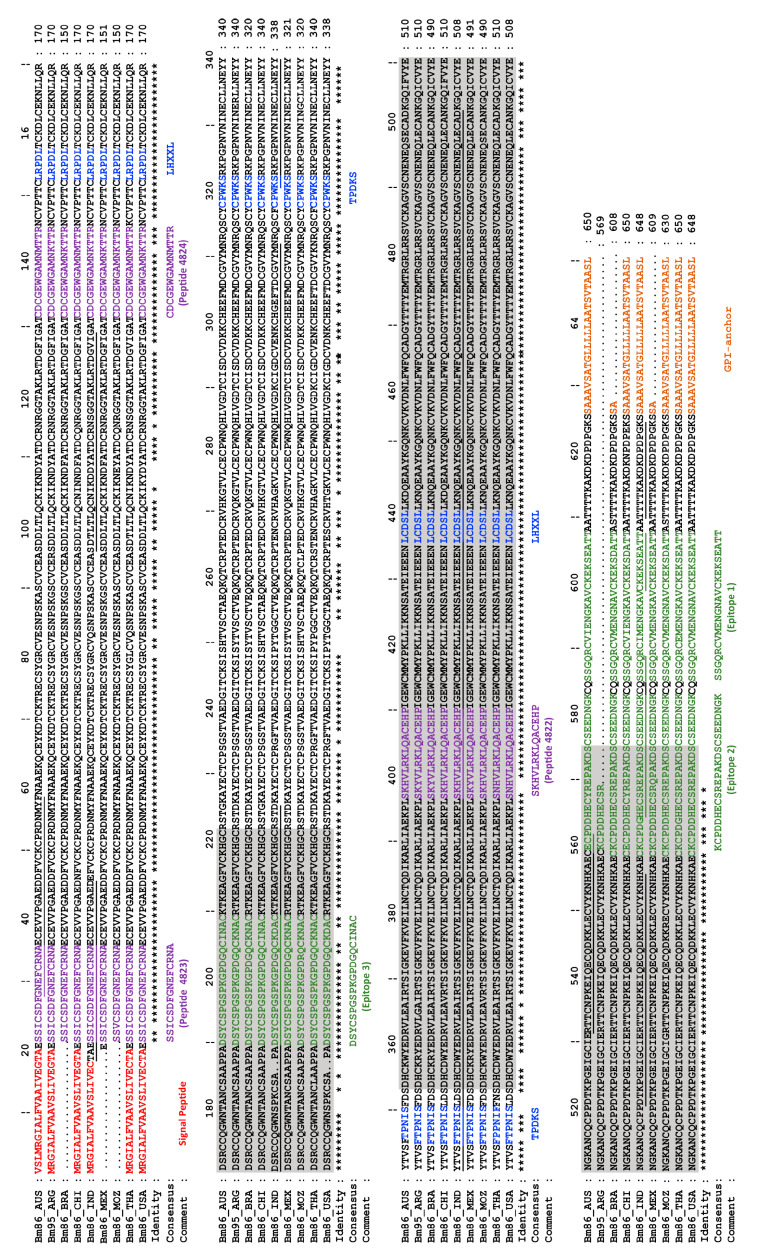
Multiple sequence alignments showing the antigenic regions (predicted and/or tested) of the Bm86 antigen isolated from *Rhipicephalus* (*B.*) *australis* (Yeerongpilly) and other geographical strains of *R. microplus*. Indicated is a standard multiple sequence alignment performed with Muscle (EMBL-EBI, https://www.ebi.ac.uk/Tools/msa/muscle/ accessed 10 June 2023), using the Bm86 sequence(s) available on the GenBank protein database (https://www.ncbi.nlm.nih.gov/protein/ accessed 10 June 2023) for the: Australian (AUS) TickGARD^®^ antigen (Yeerongpilly strain, accession #: AAA30098), Argentinian (ARG) antigen used in Gavac^®^ Plus (A strain, ‘Bm95′, accession #: AAD38381), Brazilian (Campo Grande strain, accession #: ACA57829), Chinese (CHI) (XJNJ strain, accession #: CHQBQ88489) Indian (IND) (IVRI-I strain, accession #: QFR98312), Mozambique (MOZ) (IVRI-I strain, accession #: ACZ55133), Mexico (MEX) (susceptible strain, accession #: ACR19243), Thailand (THI) (M1 strain, accession #: AJE29925), and the United States (USA) (Hidalgo 1 strain, accession #: ADQ19687). Identical amino acids that are shared between all sequences are indicated by asterisk (*) below the alignment. The predicted N- and C-terminal signal peptide(s) and GPI-anchor(s) are indicated in red and orange, respectively. Regions highlighted in grey correspond to “clone 3” and “clone 5” of *E. coli* producing truncated segments of the N- (Pos. 31- 223) and C-terminal (Pos. 351- 559) of Bm86, respectively, by Tellam et al. [49]. When tested in cattle infestation trials, the clone 3 truncation afforded a 53% reduction in female reproductive potential, and 65% was achieved for clone 5 [49]. Three peptide sequences (purple) were predicted by Patarroyo et al. [84] (i.e., 4822, 4823, and 4824), and synthetic peptides were produced, of which SBm7462^®^, consisting of all three peptides (in this order: 4822-4824-4823), produced between 72.4% and 81.05% vaccine efficacy. Additional potential epitopes were predicted by Blecha et al. [106] (green) and Parthasarathi et al. [107] (Underlined in the Indian Bm89 sequence), as well as two shared linear B cell epitopes (underlined in the Australian Bm86 sequence) between *R. microplus* and *R. decoloratus* predicted by Odongo et al. [105]. Indicated in dark blue are mimotope regions that were identified using consensus sequences determined by Prudencio et al. [108].

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
