# Peer review of "More than Three Decades of Bm86: What We Know and Where to Go"

_pathogens, 2023, doi:10.3390/pathogens12091071_

Round 1

Reviewer 1 Report

This review presents a detailed description of the use of R. microplus and R. australis Bm86 and homologues as antigens in anti-tick vaccines. The article is well written and covers from the discovery of Bm86 and its use in vaccines to recent advances in vaccinology to make this antigen more effective. Below are some suggestions and questions to improve the text.

Please correct these spelling errors:

Line 11: “form” to “from”.

Line 35: please, remove the comma.

Line 39: “on in”?

Line 63: “learnt” to “learned”

Line 122: “ausStralis”

Line 83: The acronym for Acquired tick resistance (ATR) was already explained in line 76.

Line 89: I think is missing the verb in this phrase. “Roberts and Kerr OBSERVED that a degree…”

Line 110: Rabbits are animals of the order Lagomorpha and family Leporidae. They are not rodents.

Line 221: Please, make clear what means “60% reduction in the number of acaricide treatments”.

Lines 223 to 227. The type 2 immune response is characterize by IgE, but IgG (type 1) is also produced during BM86 vaccination. How can be these two responses be characteristic of Bm86 vaccination? Otherwise, it will be interesting discuss the differences between natural tick infestation and Bm86 vaccination regarding the characteristic immune response developed and the deleterious effects in the tick larvae, nymphs and adults.

In page 21 (there are no lines in this page), is sad that a multi antigen vaccine against R. microplus was performed in tick infestation in both rabbits and cattle, where no enhanced protection was observed. However, this study shows only the cattle vaccination trial against R. microplus. Please, check the results from this work and adjust in the text.

Reviewer 2 Report

The manuscript is a review about the works developed with the Bm86 antigen, which constituted a revolution in the vaccine development against ticks and thirty years after its  discovery continues to be the gold standard of an anti-tick vaccine. This antigen is the active pharmaceutical ingredient of the only commercially available vaccines which have been applied in more than 3 million of animals in field conditions.

The topic is relevant in the field. However, despite this antigen has been used to immunize cattle for more than thirty years, its biological function in the tick gut membrane cells and the causes for the dissimilar protection against ticks have not been completely elucidated. In addition, the manuscript points the limited immune markers that were investigated across all studies for Bm86 and the consequent gaps in the understanding of the host immune system and the identification of key components and mechanisms involved in mounting a protective immune response against ticks which remain essential for designing effective anti-tick vaccines.

It is an useful review where many questions about anti-tick vaccines are addressed and it could be a reference literature for researchers working in the development of vaccines against parasites.

The authors mention in the manuscript  that “the development of  assembled genomes for R. microplus, R. annulatus and I. scapularis together DNA microarrays for ticks allowed for the downstream implementation of efficient reverse vaccinology approaches to rapidly screen for and identify protective tick antigens and/or to better elucidate functionality of existing antigens”, however, there is no anti-tick antigen that has come out applying this technology that has advanced beyond a laboratory test. Please elaborate.

It is a very good review about the works developed with the Bm86 as anti-tick antigen.

I only suggest to use "anti-tick vaccines" instead "tick vaccine" because this term expresses better the idea. This should be reviewed throughout the manuscript.

Reviewer 3 Report

Some general comments include: The review is broader than discussion about Bm86 only which the title suggests, this reviewer will suggest sections to edit or delete accordingly. The review needs to be more specific. There are some examples of either misinterpretation of cited papers or the authors used citations of citations from other publications without checking the original text themselves. Note the manuscript is approximately 4000 words over the limit of 12,000 words according to MDPI Pathogens instructions (personal communication to reviewer)

A detailed list of comments and edits are provided here:

1) Note Bm86 was discovered in '1986', that is why it is called Bm'86' - unlike when the authors claim which is 'late 1980s' or a 1988 reference in other sections - for the abstract - use '1986' instead. Bm86 was actually discovered by Dr Dave H Kemp and colleagues (you will note if you look at his publication history that he has about 15 years of research on B. microplus leading to this discovery - thus it is not trivial) - the correct references for the discovery is: Immunization of cattle against Boophilus microplus using extracts derived from adult female ticks: feeding and survival of the parasite on vaccinated cattle - PubMed (nih.gov). (you do not have this reference). Immunization of cattle against Boophilus microplus using extracts derived from adult female ticks: Histopathology of ticks feeding on vaccinated cattle - ScienceDirect (your reference #40). These papers form the seminal work for the discovery of Bm86 and also 'partial amino acid sequencing of Bm86 was obtained in 1986' as reviewed by Willadsen in your listed reference #63 - which enabled the team to then identify the rest of the gene for Bm86. Willadsen subsequently led the commercal aspect of the research which eventually became TickGARD. Please also edit lines 146-148 noting the '1986' discovery and referencing according this point. 

2) The title 'Three decades...' from 1986 - 2023 is 37 years - thus consider new title 'More than three decades....' or 'Almost four decades...'

3) Reference #1 (de la Fuente and Kocan 2003) is the apparent source of ancient tick and TBDs from ancient Egypt and ancient Greece however reference #1 makes these comments without citing any sources. It is thus probably not the best reference. Try to find 'real sources' for citations to demonstrate optimal academic integrity, an example is Huchet et al. 2013 https://doi.org/10.1016/j.ijpp.2013.07.001 - you may find others if you look harder or any relevant historical texts. Ticks are in fact are much much much older found in 99million year old amber with a tick from feathered dinosaur fossil, see: Penalver et al 2017 DOI: 10.1038/s41467-017-01550-z Edit this historical part (line 23) accordingly. 

4) Figure 1 and Supplementary Table S1 should be merged. This is not a review paper describing the chronological order of molecular technological discoveries but it is meant to be about Bm86 and ticks. Thus merge the research milestones (tick vaccines and acaricides) and perhaps tag the most relevant technology milestones which for tick research the relevance is probably for chemicals, recombinant expression (for Bm86 expression), and DNA sequencing milestones (the latter for tick genomes) as 'most relevant'. Delete unnecessary vaccine discoveries not relevant to this review (ie. discovery of subunit vaccines is relevant to this review).  Also add 1986 - Bm86 discovered through partial amino acid sequencing. 

5) Lines 56, 124 'CSIRO' (not CIRO)

6) lines 68-70 - the Australian tick has a longer history than this - using your own reference #20 (1918) you will see it was originally called 'Boophilus australis' and as such has recently reverted to its original name of 'australis'

7) Reference #20 discusses B. australis infestations on farms in 'Queensland' Australia (not 'New South Wales' - line #74). This reviewer is convinced that this 1918 article perhaps was not read by the authors - the correct wording would be from the conclusions of this publication which states 'Many cattle become habituated to tick infestation and this, in individual cases leads to some degree of resistance'- note that the studies in this publication were all on Bos taurus breeds where full resistance is not entirely possible. Another statement from this study 'Tick resistant cattle are known from a number of Queensland localities. Asiatic breeds are tick resistant. Of the various breeds commonly met with in this State, Jerseys appear to be less affected than others by ticks.' Thus please update your general comment according to the conclusions published in reference #20. Also line 73 writes 'In 1918, Johnston and Bancroft(20) which is correct but the reference list has 'Johnston and Mackerras'. Reference #20 should be: Johnston TH; Bancroft MJ. A tick resistant condition in cattle.  Proceedings of the Royal Society of Queensland 1918, 30, 219-317. Please correct.

8) At the start of the paragraph on line 146, the authors have not described the trials describing the use of native Bm86 as vaccine which needed many ticks to produce enough Bm86, hence they quickly moved to recombinant production (Rand et al 1989 #18) once scientifically viable. This information can be found in your reference #45 whereby 50,000 ticks were used to prepare Bm86 native antigen- it was also described in the justification of the study by Rand et al 1989. The short amino acid sequence allowed the development of primers to obtain the full Bm86 sequence to enable expression.

9) Lines 182-218. This could be truncated and the order corrected- the review by Willadson reference #63 may help. First CSIRO patented Bm86, then Cuba produced GAVAC and actually published GAVAC trials before TickGARD, then Australians made TickGARD plus in Pichia but GAVAC had already expressed Bm86 in Pichia. The order of the current text has TickGARD, tickGARDplus followed by GAVAC which is incorrect. Please correct the timeline and truncate the text.

10) Tables 1 and 2. This reviewer suggests:

a) adding information about the vaccination regime (1, 2 or 3 shots), followed by the infestation regime (1000 larvae or 10,000 larvae and for how many days) - this can vary the results considerably. This will also affect your 'immune markers' data column - you could then discuss these discrepancies. Thus column called 'adjuvant and additives' should be 'vaccination regime'

b) in the Reference column simply use numbers corresponding to patents and publications - patents are always on line documents that you can cite as a number in this table and the patent details should all be in the reference list (they are not currently)

c) instead of 'bovine species' - put 'breed' the title of the table already has 'bovine'

11) Figure 2 is 'unreadable' - suggest to present it larger on a landscape full page. 

12) line numbering stops on the page containing Figure 2 and the page numbers change from  page 11 (start of Table 2) to page 2 (follows from 11 and starts counting from 2 onwards) -Comment only - however some confusion of locations of edits from this point onwards may ensue.

13) One of the issues not really embraced by the authors is that Bm86 is just not that immunogenic - as it needs 3-4 boosts per year - it is mentioned in passing only 

14) Section 5 should be deleted as it discusses RNA interference which is a gene knockdown method and not related to vaccination or host immune responses. The fact that Bm86 knockdown does not demonstrate a phenotype proves this point! This will assist to delete over lengthy text in this review. 

15) Section 7, 2nd paragraph - TickGARDPlus was discontinued in 2010 (TickGARD was superseded by TickGARD plus - see reference #63 for the year).

16) 'Freund's' adjuvant is spelt incorrectly throughout the manuscript - please correct. 

17) Conclusion - 'After almost 4 decades of...........)

18) Summary of review - the authors need to determine sections that could be truncated - in many cases, a focus on one article at a time makes the text description lengthy - a review requires summarising several articles at once which may help to truncate the lengthy manuscript to ~12,000 words from almost 16,000 (not including tables and references). The only section this reviewer suggested to delete was the RNAi section no. 5) and also perhaps delete introduction sections about discoveries in vaccinology - keep it to Bm86 relevance only. Other text could simply be written more succinctly and less repetitively. With the areas where the authors have misinterpreted the nuances of Bm86 discovery - perhaps the authors would have benefitted by having either a Cuban or Australian Bm86 researcher as a co-author which would have avoided these misinterpretations or incorrect assumptions based on literature only. For the most part, the authors had most of the relevant references but the order of events and how Bm86 was discovered and produced as a vaccine was muddled. 

Round 2

Reviewer 3 Report

No comments.